# Cas9 is mostly orthogonal to human systems of DNA break sensing and repair

**Ekaterina A. Maltseva**[1☯], **Inna A. Vasil'eva**[1☯], **Nina A. Moor**[1], **Daria V. Kim**[1,2], **Nadezhda S. Dyrkheeva**[1], **Mikhail M. Kutuzov**[1], **Ivan P. Vokhtantsev**[1], **Lilya M. Kulishova**[1], **Dmitry O. Zharkov**[1,2]*, **Olga I. Lavrik**[1,2]*

**1** SB RAS Institute of Chemical Biology and Fundamental Medicine, Novosibirsk, Russia, **2** Novosibirsk State University, Novosibirsk, Russia

☯ These authors contributed equally to this work.
* lavrik@niboch.nsc.ru (OIL); dzharkov@niboch.nsc.ru (DOZ)

**Data Availability Statement:** All relevant data are within the paper and its Supporting Information files.

## Abstract

CRISPR/Cas9 system is  powerful gene editing tool based on the RNA-guided cleavage of target DNA. The Cas9 activity can be modulated by proteins involved in DNA damage signalling and repair due to their interaction with double- and single-strand breaks (DSB and SSB, respectively) generated by wild-type Cas9 or Cas9 nickases. Here we address the interplay between *Streptococcus pyogenes* Cas9 and key DNA repair factors, including poly (ADP-ribose) polymerase 1 (SSB/DSB sensor), its closest homolog poly(ADP-ribose) polymerase 2, Ku antigen (DSB sensor), DNA ligase I (SSB sensor), replication protein A (DNA duplex destabilizer), and Y-box binding protein 1 (RNA/DNA binding protein). None of those significantly affected Cas9 activity, while Cas9 efficiently shielded DSBs and SSBs from their sensors. Poly(ADP-ribosyl)ation of Cas9 detected for poly(ADP-ribose) polymerase 2 had no apparent effect on the activity. *In cellulo*, Cas9-dependent gene editing was independent of poly(ADP-ribose) polymerase 1. Thus, Cas9 can be regarded as an enzyme mostly orthogonal to the natural regulation of human systems of DNA break sensing and repair.

## Introduction

Enzymatic systems based on the clustered regularly interspaced palindromic repeats (CRISPR)-associated proteins, Cas9 endonuclease in particular, have recently been transformed to powerful gene editing tools. Due to the precise RNA-guided recognition of target DNA sequences they provide tremendous possibilities for specific genome, epigenome and transcriptome manipulations and thus promise fast progress in medicine, agriculture and basic research techniques [1–5].

The molecular mechanism underlying editing by Cas9 includes recognition of a short DNA sequence (protospacer-adjacent motif, PAM) by the C-terminal domain of the protein, subsequent separation of DNA strands upstream of the PAM and annealing of the guide RNA to its complementary target DNA [3] (here and below we use the name Cas9 to refer to the protein from *Streptococcus pyogenes*, by far the most extensively characterized member of type II Cas endonucleases). If all these steps are successfully accomplished, the target DNA is cleaved by

**Funding:** This research was supported by Russian Science Foundation (grant 21-64-00017). Partial salary support from the Russian Ministry of Science and Higher Education (State funded budget project 121031300056-8 to D.O.Z.) is acknowledged. The funders had no role in study design, data collection and analysis, decision to publish, or preparation of the manuscript.

**Competing interests:** The authors have declared that no competing interests exist.

cooperative action of RuvC and HNH domains of Cas9 between −4 and −3 positions relative to PAM thus generating a blunt-ended double-strand break (DSB) [3, 6]. DSBs are perceived by the cell as highly toxic DNA lesions leading to genome instability, and the site-specific breaks made by Cas9 are subject to repair via either the accurate homologous recombination (HR) pathway or several error-prone mechanisms such as classical non-homologous end joining (c-NHEJ), microhomology-mediated end joining, or single-strand annealing [7, 8]. These pathways, while partially overlapping, differ in the major participating proteins and modes of damage detection.

Poly(ADP-ribose) polymerase 1 (PARP1) is one of the key regulators of DNA damage response in human cells [9, 10]. It is a highly abundant multidomain enzyme that, when activated by binding to a single-strand break (SSB) or DSB, catalyses $NAD^+$-dependent assembly of long branched poly(ADP-ribose) (PAR) chains (PARylation) on various molecular targets, including histones, chromatin structure modulators, DNA repair enzymes, and PARP1 itself. PARylation represents dynamic posttranslational modification, which promotes signal transduction and regulates the choice of DNA repair pathway. PARylated proteins may dissociate from DNA exposing the break for further processing, or undergo liquid/liquid phase separation forming membraneless compartments where the repair could take place [11, 12]. Although the role of PARP1 is best documented in SSB repair, there is increasing evidence for PARP1-mediated modulation of c-NHEJ efficiency and for PARP1 participation in HR and alternative end-joining pathways of DSB repair [13–16]. PAR synthesis can be catalysed by several other PARP homologs, of which poly(ADP-ribose) polymerase 2 (PARP2) is the most closely related to PARP1 [10, 17]. Consistent with this overlapping catalytic proficiency, PARP2 is epistatic with PARP1, with single knockouts viable but the double knockout lethal in a mouse model [18].

As Cas9 is a totally foreign protein for human cells, it is unclear how it would interact with the cellular DNA damage response and DNA repair systems, which have most likely evolved to protect cells from DSBs induced by radiation or chemical DNA damage and replication failures. Cas9 produces a DSB but, at least *in vitro*, releases this product very slowly [19, 20]. The cellular factors that might facilitate the enzyme turnover and expose the nascent DSB are poorly known; Cas9 holds on DNA so tightly that it could hardly be displaced even by replication and transcription machinery [21–23]. Two ways that have found some experimental support so far involve FACT, a nucleosome disassembly factor that normally operates on H2A/H2B histones removing them from chromatin, and ubiquitylation or sumoylation followed by proteolysis [24, 25]. Even with this, the exact mechanism of Cas9 removal has not been addressed, and many other chromatin remodelling factors could be involved. It is feasible that PARP1 or PARP2 could modify intracellular Cas9, however, this possibility has not been addressed experimentally so far. Moreover, Cas9 associated with a DSB could shield the ends of the break from DNA damage-signalling proteins.

Along with PARPs, other proteins that co-operate in DNA breaks recognition and repair could be involved in the events following the cleavage by Cas9. For example, the Ku antigen (Ku70/80, a heterodimer of Ku70 and Ku80 polypeptides) is the primary DSB recognition factor in the c-NHEJ pathway [26] and thus might play a role of a factor that displaces Cas9 from the DSB product. DNA ligase I (LigI), an SSB sensor, is involved in the final step of DNA repair, the nick ligation [27, 28], and, in principle, could compete with Cas9 nickase mutants (nCas9 D10A and nCas9 H840A) for their cleavage products. Replication protein A (RPA) involved in DNA replication, recombination, and repair is responsible for the stabilization of single-stranded DNA stretches [29, 30]. RPA is able to unwind the DNA duplex and could potentially facilitate the annealing of guide RNA to the protospacer in DNA. Y-box-binding protein 1 (YB1) participates in DNA repair and multiple mRNA-dependent processes [31, 32]

and is potentially able to compete with Cas9 for sgRNA interaction. Thus, possible interplay between Cas9 and DNA repair factors, including PARPs, LigI, Ku70/80, RPA, and YB1, in the process of genome editing is worth attention.

## Results

### Cas9 retains its activity in the presence of PARP1 and PARP2

DNA break sensor proteins PARP1 and PARP2 synthesize PAR from $NAD^+$ when activated upon binding damaged DNA [9, 33]. We have investigated whether human PARP1 or murine PARP2 affect the activity of Cas9/sgRNA in the absence and in the presence of $NAD^+$. Human and murine PARP2 are highly homologous [34] and are expected to be indistinguishable in most functional aspects. At first, an oligonucleotide duplex containing the well-characterized Sp2 protospacer [6] and a TGG PAM was used as a substrate (S1 Fig and S1 Table). In the duplex, either the target strand (dsDNA1/2*) or the non-target (dsDNA1*/2) strand was $^{32}$P-labelled to distinguish the effect of PARPs on the activity of HNH- and RuvC-like nuclease domains, respectively. Importantly, to ensure access of PARPs to DNA, we used an equimolar ratio of the substrate, Cas9 protein, and sgRNA, rather than a large excess of Cas9/sgRNA often used in functional studies of Cas9. Under these conditions, Cas9 cleaved 34 – 42% of the dsDNA1/2 substrate in 30 min. No significant change in the cleavage efficiency of either strand was observed even at a high excess of either PARP over Cas9, regardless of the presence of $NAD^+$ (Fig 1A and 1B). To exclude PARP1/PARP2 activation at the duplex ends, we then have used the supercoiled pLK1 plasmid as a substrate and compared its cleavage by Cas9 and its nickase mutants (nCas9 D10A and nCas9 H840A) in the absence and presence of PARP1/ PARP2, without or with $NAD^+$. Under the experimental conditions used, both nicked (SSB-containing, P1) and linear (DSB-containing, P2) products accumulated (Fig 1C and 1D). It is known that PARP1 and PARP2 are activated in the presence of such DNA intermediates with different efficiency [35, 36]. Wild-type Cas9 demonstrated identical cleavage rates in the presence or in the absence of PARP1 or PARP2 (Fig 1C and 1D). Despite PARP1 and PARP2 are primary SSB sensors in human cells and induce an immediate response to DNA damage by PAR synthesis, we also have observed no impact of the PARPs (unmodified and automodified) on the activity of Cas9 nickase mutants (S2 and S3 Figs).

### Cas9 and PARPs bind DNA substrates independently

To explore possible reasons under the absence of PARP1/PARP2 impact on the Cas9 activity, we compared binding of these enzymes to DNA. We estimated the affinity of PARPs, Cas9 and its mutants (nCas9 D10A, nCas9 H840A and dCas9, the double D10A/H840A mutant fully inactive in DNA cleavage) for $^{32}$P-labelled dsDNA1/2* using the electrophoretic mobility shift assay (EMSA) (S4 and S5 Figs). Unlike free Cas9, the preformed Cas9/sgRNA complex efficiently bound dsDNA1/2, mostly forming a well-resolved ternary complex (Cas9/sgRNA-dsDNA) (S4A Fig). Apparent equilibrium dissociation constants ($K_d$) of the complexes were approximated as the effective concentrations of Cas9/sgRNA at the half-maximal extent of DNA binding ($EC_{50}$). The $K_d$ value for Cas9/sgRNA was 7.7 ± 0.4 nM (S4D Fig). The inactivating single and double mutations in Cas9 did not significantly change the affinity (S4B–S4D Fig). Compared with Cas9/sgRNA, both PARP1 and PARP2 had much weaker apparent affinity for DNA ($K_d$ = 53 ± 5 nM and 200 ± 16 nM, respectively; S5A–S5C Fig). It should be noted, however, that under the conditions used PARP1 and PARP2 are positively charged and the PARP–DNA complexes enter the gel rather inefficiently; therefore, the amount of PARP-bound DNA was estimated from a decrease in the amount of free DNA. The addition of $NAD^+$ resulted in disappearance of PARP–DNA complexes and release of free DNA,

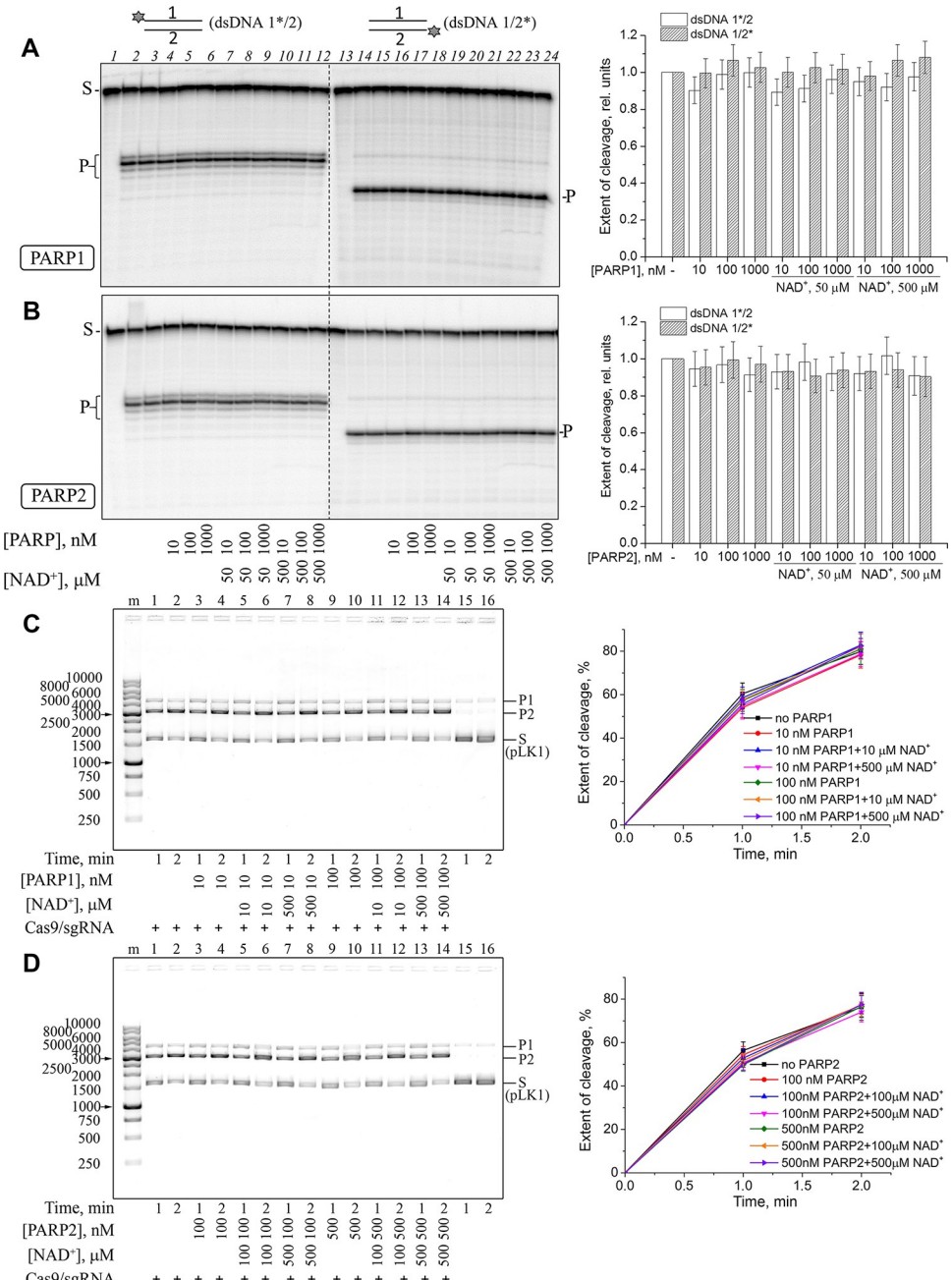

**Fig 1. Cleavage of the oligonucleotide or the plasmid substrates by Cas9/sgRNA in the presence of PARP1 and PARP2.** Cas9/sgRNA (10 nM) was incubated with $^{32}$P-labelled dsDNA1*/2 or dsDNA1/2* (10 nM; A, B) or with pLK1 DNA (10 ng/µl; C, D) and the indicated amounts of NAD$^+$, PARP1 (A, C) or PARP2 (B, D). S, substrate (dsDNA or supercoiled plasmid); P, dsDNA cleavage product; P1, SSB-containing product (nicked plasmid); P2, DSB-containing product (linear plasmid). The sizes of DNA markers are shown next to the gel images. Bar charts (A, B) show the dsDNA cleavage in the presence of PARP1 or PARP2 normalized to the cleavage in the absence of PARPs (the mean ± SD, $n = 3$). The curves (C, D) show the accumulation of P1+P2 products of pLK1 cleavage under the indicated conditions (the mean ± SD, $n = 3$).

confirming that the automodification of PARPs significantly reduced their affinities for dsDNA1/2 (S5A and S5B Fig).

Having characterized binding of the individual proteins, we then inquired whether PARP1 or PARP2 influences Cas9/sgRNA binding to dsDNA1/2. The Cas9/sgRNA–DNA complex disappeared and low mobility complexes (LMC) accumulated under the gel wells in the presence of PARP1/PARP2 taken in excess (Fig 2A and 2B). However, PARP1/PARP2 form similar complexes alone, and the LMCs disappeared under PAR synthesis conditions whereas the Cas9-DNA complexes remained mostly unaffected. These results, together with the lack of significant PARPs effects on the Cas9 activity (see above) strongly suggest that even if Cas9/sgRNA–PARP–DNA complexes form, PARPs and Cas9 interact with different parts of dsDNA1/2.

To exclude binding of PARPs to the DNA duplex blunt ends, we further utilized the supercoiled pLK1 plasmid as a DNA ligand. Binding of dCas9 and its complex with sgRNA to the plasmid was assayed in the presence of $Mg^{2+}$, with or without PARP1/PARP2 (Fig 2C and S6 Fig). While the addition of increasing concentrations of free dCas9 induced a shift of the band of plasmid DNA, dCas9/sgRNA had no effect (Fig 2C, lanes 14–16 vs. lanes 17–19). The dCas-induced band shift evidently reflects nonspecific binding of many protein molecules at multiple sites in the plasmid. Previously, Cas9 was shown to bind long DNA in a sequence-independent manner, although more than two orders of magnitude weaker in comparison with the site-specific binding of Cas9/sgRNA complex to its target DNA [19]. On the other hand, the specific binding of dCas9/sgRNA to a single site contributes little (~190 kDa overall) to the total molecular weight of the plasmid (~3000 kDa), causing a negligible band shift. Nonspecific pLK1 binding by PARP1 and PARP2 was detected as a band shift at 100 nM and 500 nM, respectively (S6 Fig), reflecting higher affinity of PARP1 for the undamaged DNA structure. Indeed, PARP1 was shown to exceed PARP2 by ~5-fold in the strength of nonspecific interaction with different DNAs [37]. The PARP1-induced band shift was not affected by addition of sgRNA but the band was supershifted to different extents upon addition of dCas9/sgRNA or dCas9 (Fig 2C), suggesting simultaneous interaction of the proteins with plasmid DNA. These results corroborate the hypothesis that PARPs do not interfere with the formation of a productive Cas9/sgRNA-DNA complex.

## Cas9 binds PAR

The presence of DNA- and RNA-binding domains in the Cas9 protein suggests that it might also interact with the negatively charged PAR polymer. This possibility was explored by EMSA (Fig 3A). The bulk [$^{32}$P]PAR synthesized by PARP1 was used without fractionation. $K_d$ values were approximated as $EC_{50}$ of the proteins at the half-maximal extent of PAR binding. The binding parameters summarized in Fig 3C show that Cas9, both nCas9 variants, and dCas9 interact with PAR with similar affinities, which are two orders of magnitude lower than the affinities of sgRNA-bound wild-type and mutant Cas9 forms for dsDNA substrate (S4D Fig). We also explored the influence of sgRNA on the binding of Cas9 and dCas9 to PAR. Addition of an equimolar amount of sgRNA significantly reduced the extent of PAR binding by Cas9/dCas9, as evidenced by 88–93% release of free PAR (Fig 3B). These results are consistent with a much stronger interaction of Cas9/dCas9 with sgRNA than with PAR. The competition between sgRNA and PAR for Cas9 may either be direct competition for the RNA-binding domain, or result from conformational rearrangement of the protein bound to sgRNA [38, 39].

## Cas9 is PARylated by PARP2 *in vitro*

To further follow possible interplay between Cas9 and PARPs, we inquired whether PARP1 or PARP2 could PARylate Cas9. The dsDNA substrate of Cas9 was used to activate PARPs. Since

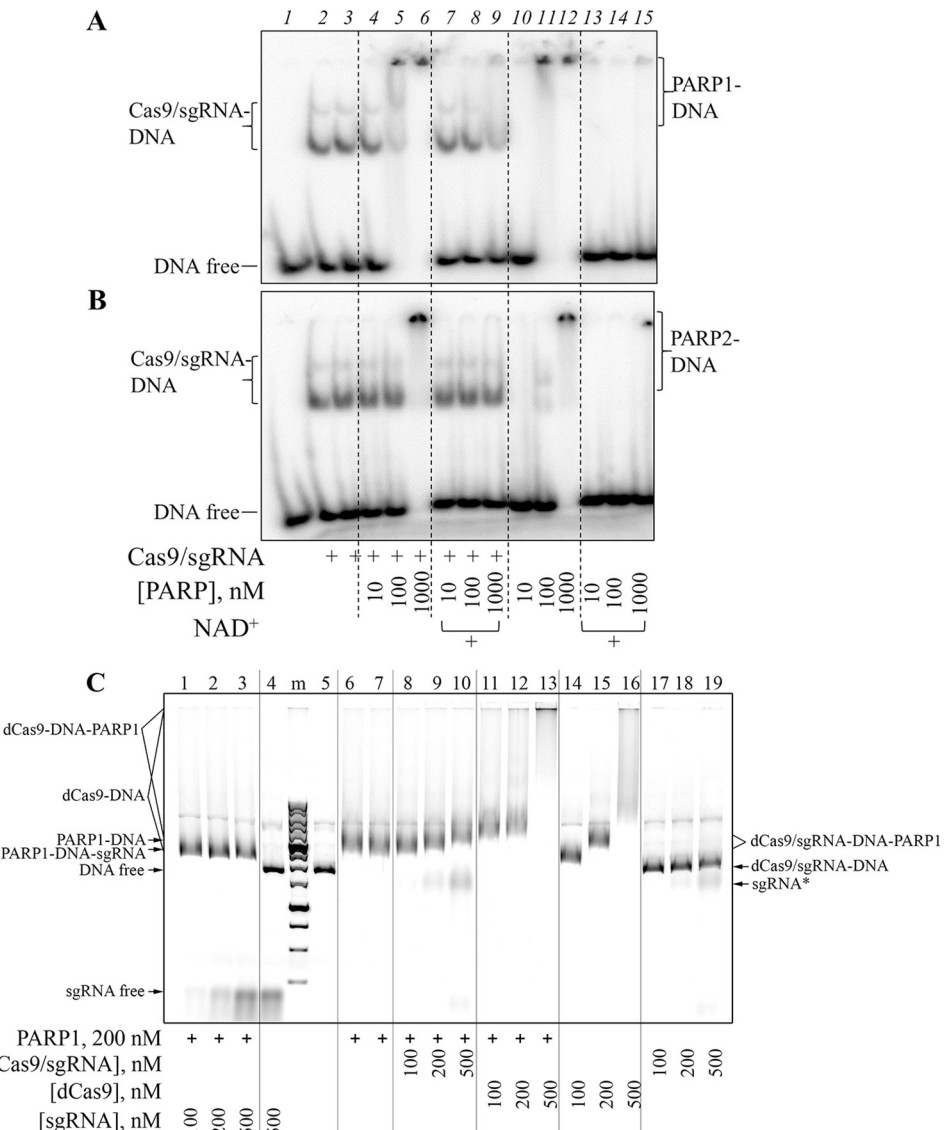

**Fig 2. Cas9/sgRNA, PARP1 and PARP2 binding to DNA.** dsDNA1/2* (10 nM) was incubated with Cas9/sgRNA (10 nM) in the absence or in the presence of PARP1 (A) or PARP2 (B), without or with 500 μM $NAD^+$, and analysed by EMSA. In Panel C, pLK1 (10 ng/μl) was incubated with the indicated amounts of dCas9/sgRNA, dCas9 alone or sgRNA (in the presence of $Mg^{2+}$), in the absence or in the presence of PARP1 (200 nM). Positions of free (unbound) DNA or sgRNA and of various complexes are shown next to the respective gel images.

the Cas9-induced DNA cleavage occurs only in the effector complex with the guide RNA, we tested the ADP-ribosylation activity of PARPs in the presence of sgRNA and dsDNA added separately or together in an equimolar ratio. It was shown previously that PARP1 and PARP2 automodified at high $NAD^+$ concentrations form large associates that do not enter polyacrylamide gels due to intermolecular bridging of PAR chains by $Mg^{2+}$ [40, 41]. Therefore, the heteromodification of Cas9 protein was explored at low $NAD^+$ and defined PARP concentrations or by addition of protein factors regulating the PAR chain length [42, 43]. Since Cas9 (158.4 kDa) migrates through the SDS-PAG slower than PARP1 (113 kDa) and PARP2 (68 kDa), the auto- and heteromodification products may overlap. For this reason, the total ADP-ribose

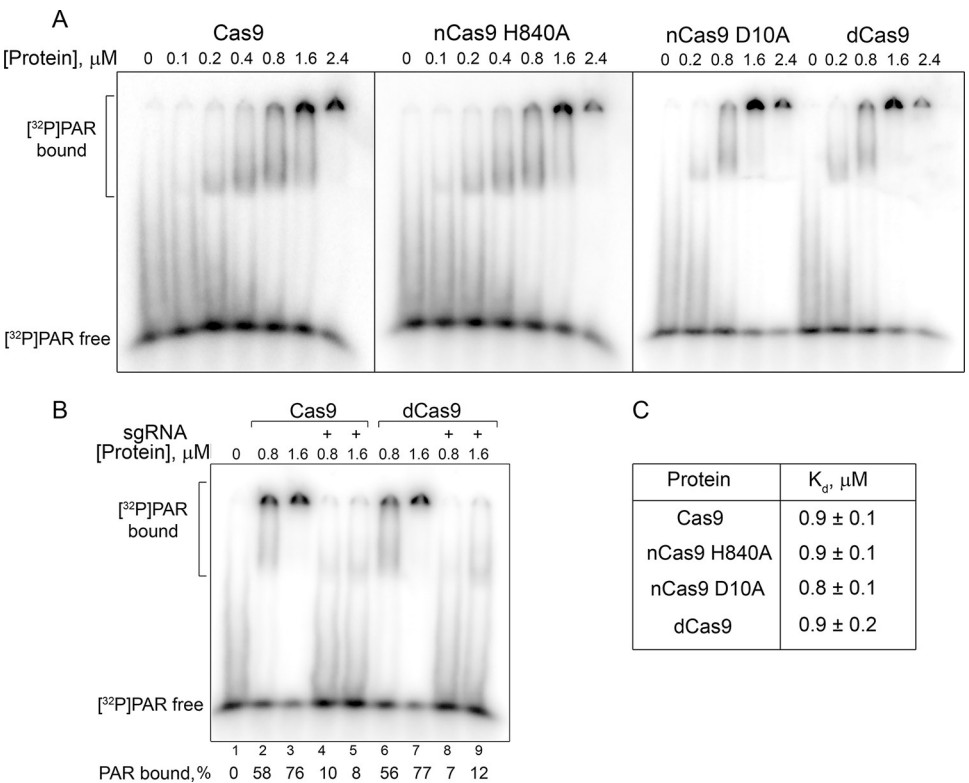

**Fig 3. PAR binding by Cas9.** $^{32}$P-labelled PAR was incubated with Cas9, nCas9 H840A, nCas9 D10A, and dCas9 apo-proteins (A) or Cas9/dCas9 1:1 complexes with sgRNA (B). The complexes were separated from free PAR by 5% native PAGE and quantified as described in Methods. Since PAR complexes with proteins poorly enter the gel and may be partly lost to the running buffer, the fraction of bound PAR was estimated from the decrease in the amount of unbound PAR compared to the protein-free sample. (C) Apparent dissociation constants of the complexes (the mean ± SD, $n$ = 3).

amount covalently bound to the proteins was quantified. The reaction conditions were optimized (data not presented) by using XRCC1 as a known target of PARylation and a negative regulator of PAR chain elongation [42, 44, 45]. Using a PARP1 mutant G972R, which synthesizes short hypobranched PAR chains, clear-cut bands of ADP-ribosylated PARP1 can be obtained [46]. The results of PARP1-catalysed modification are shown in Fig 4A. No appreciable sgRNA-dependent activation of PARP1, in the absence and presence of Cas9 was detected. The total yield of PARP1 autoPARylation in the presence of dsDNA (or dsDNA with sgRNA) was increased slightly (~18%) upon addition of Cas9, whereas no Cas9 PARylation was evident.

Under the same reaction conditions, the dsDNA-dependent activity of PARP2 was significantly weaker in comparison with that of PARP1 (Fig 4A and 4B). Nonetheless, covalent labelling of Cas9 with [$^{32}$P]ADP-ribose induced by dsDNA, sgRNA or their mixture was detected in the PARP2-catalysed reaction (Fig 4B). The PARP2 automodification levels in the presence of dsDNA, sgRNA or their mixture were only slightly different from the level of PARP2 basal (DNA-independent) activity. When Cas9 was added, the PARP2-catalysed protein ADP-ribosylation level increased significantly (2–6.4-fold). The new band labelled in the presence of Cas9 corresponded to the mobility of Cas9 and migrated much slower than the automodified PARP2. The highest level of Cas9 ADP-ribosylation in the presence of dsDNA and sgRNA suggests that Cas9/sgRNA bound to dsDNA is a preferable target for PARP2.

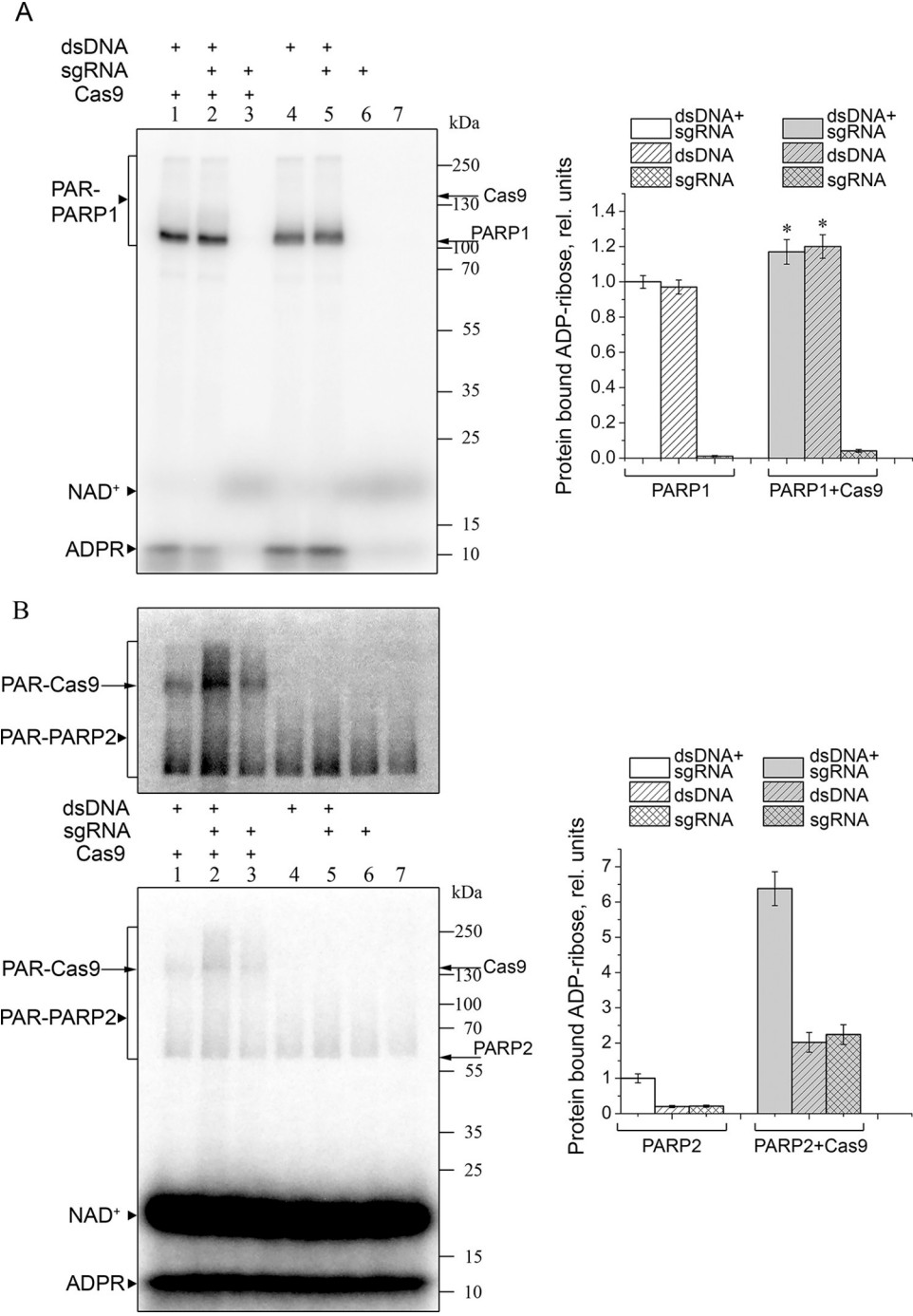

**Fig 4.** Poly(ADP-ribosyl)ation catalysed by PARP1 G972R (A) and PARP2 (B) in the absence and in the presence of Cas9. PARP1 G972R (300 nM) or PARP2 (300 nM) was incubated for 30 min with [$^{32}$P]NAD$^+$ (0.8 μM) and dsDNA, sgRNA and Cas9 (1 μM each) as indicated. The positions of $^{32}$P-PARylated PARP1 (PAR-PARP1), PARP2 (PAR-PARP2) and Cas9 (PAR-Cas9) are marked to the left, and those of native proteins and protein molecular weight markers, to the right of the gel images. The top image in panel B shows the same gel after a longer exposure. Bar charts show relative levels (the mean ± SD, $n = 3$) of total PARylated protein normalized for the level of modification in the presence of dsDNA and sgRNA without Cas9. A slight Cas9-induced increase in the level of PARP1-catalysed PARylation (A) was statistically significant p ≤ 0.05 (*).

## Cas9-generated DSBs are shielded from PARP1 binding

Cas9 is known to release its DSB cleavage product very slowly [19, 20]. To explore the activation of PARP1 by Cas9-generated DSBs, we have compared the PARP1 automodification in the presence of pLK1 plasmid DNA, either supercoiled or linearized by Cas9/sgRNA or EcoRI restriction endonuclease (Fig 5). In the presence of pLK1 pre-treated with Cas9/sgRNA followed by heating to release Cas9 from the cleavage products, PARP1 automodification was 1.7-fold higher (after a 30-min incubation) than in the presence of supercoiled pLK1. Moreover, it was comparable with the PARylation level detected in the presence of EcoRI-linearized pLK1. Thus, deproteinized Cas9-generated DSBs can stimulate PARP1. At the same time, the level of PARP1 automodification after a 30-min incubation with Cas9/sgRNA without heating was only ~1.2-fold higher compared to the incubation without Cas9. Thus, Cas9 can shield its cleavage product from binding and subsequent activation of PARP1.

PARPs could affect Cas9 activity indirectly through their interaction with DNA repair proteins and various DNA-binding factors. We have compared the resistance of Cas9-generated cleavage product to degradation by HEK293 and HEK293 $PARP1^{-/-}$ cell extracts using $^{32}$P-labelled dsDNA1/2 as the substrate. The yield of non-target DNA strand cleavage product decreased significantly in the presence of both cell extracts (6.7-fold and 7.4-fold, respectively, at the highest protein concentration used; S7A, top, and S7B Fig). At the same time, fast-migrating bands appeared in the gel, probably corresponding to degradation products of the labelled non-target strand. In contrast, the yield of target DNA strand cleavage product decreased only slightly (1.4-fold for HEK293 and 1.6-fold for HEK293 $PARP1^{-/-}$ with 4 μg of the extract; S7A, bottom, and S7B Fig). Thus, Cas9 protects its cleavage product from subsequent degradation by other proteins in an asymmetric manner, maintaining stronger contacts with the target DNA strand hybridized with sgRNA. This result is consistent with previously published data shown that the 5′-terminal part of the protospacer region of the non-target strand in the Cas9/sgRNA complex is accessible for cleavage by P1 nuclease and DNA glycosylases UDG and SMUG1 [38, 47]. The observed DNA protection by Cas9 is most likely independent of PARP1.

## Cas9 retains its activity in the presence of other SSB and DSB repair sensors

Eukaryotic DNA ligases I, III and IV are important DNA repair enzymes catalysing, with different efficiencies, the ligation of DNAs containing SSBs or DSBs [48]. LigI can potentially modulate the nCas9-induced cleavage due to its high affinity for SSBs. We have explored the influence of human LigI on DNA cleavage by Cas9 nickases, using the pLK1 plasmid as a substrate to prevent additional low-affinity interaction of LigI with blunt DNA ends. The presence of a 40-fold molar excess of LigI in the cleavage reaction mixture had no significant effect on the initial rate and maximal extent of the non-target strand cleavage by either nCas9 H840A/sgRNA or nCas9 D10A/sgRNA (Fig 6A and S8 Fig). The cleavage product generated by either nCas9 was nearly completely ligated by LigI added after thermal inactivation of the nickase (Fig 6B). However, the amount of cleaved DNA in the samples not subjected to heat treatment remained unchanged upon the addition of LigI. Hence, stable binding of either nCas9 to its SSB product prevents the processing of SSB by LigI.

The main product of Cas9-catalysed DNA cleavage is a DSB. The major DSB repair pathway, except during the S and G2/M phases of the cell cycle, is the c-NHEJ, where the primary damage sensor is the Ku antigen (Ku70/80) [26]. To test whether Ku can displace Cas9 from its DSB product for subsequent repair, we have explored the effect of human Ku on the Cas9-catalysed cleavage of the pLK1 plasmid (again to avoid any interaction of Ku with DNA blunt

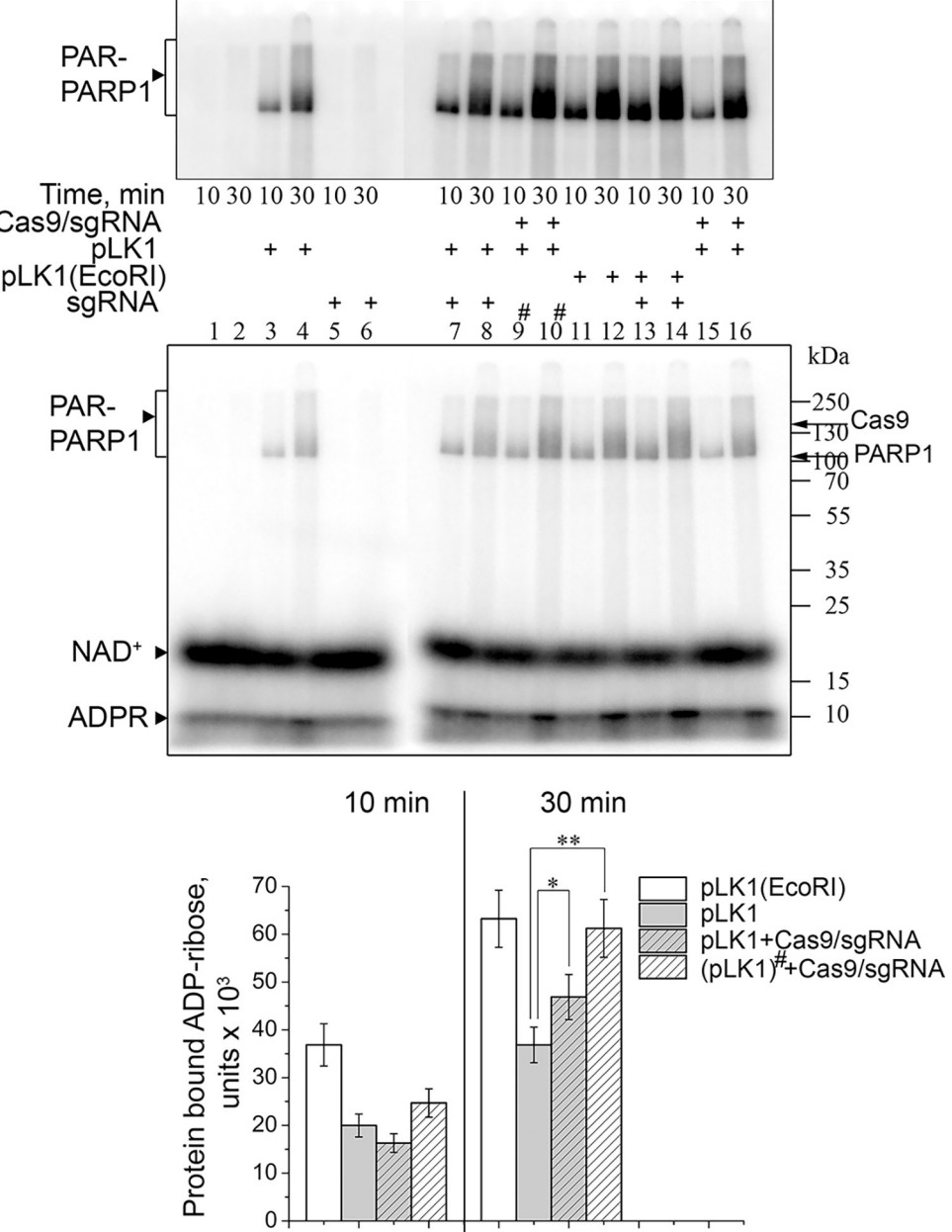

**Fig 5. Cas9 protects its cleavage product from binding and activation of PARP1.** PARP1 (200 nM) was incubated for 10 or 30 min with [$^{32}$P]NAD$^+$ (1 µM) and supercoiled or EcoRI-linearized pLK1 plasmid (100 ng/µl), with Cas9/sgRNA (100 nM) added as indicated. Samples 9 and 10 (marked with an asterisk) were pre-incubated for 20 min with Cas9 followed by thermal inactivation (70˚C for 5 min) before addition of PARP1 and NAD$^+$. The position of $^{32}$P-PARylated PARP1 is marked to the left, and those of native proteins and protein molecular weight markers, to the right of the gel images. The top image shows the same gel after a longer exposure. Bar charts show the amount of protein-bound ADP-ribose (the mean ± SD, $n = 3$) produced under various reaction conditions as specified in the legend. Statistically significant differences in the level of PARylation discussed in the text are marked $p \leq 0.05$ (*), $p \leq 0.002$ (**).

ends) (Fig 6C). No significant effect was found either with simultaneous addition of Cas9/sgRNA and Ku (in a 20-fold excess over Cas9) or with pre-incubation of the latter with DNA. The DNA-binding activity of Ku was confirmed by its incubation with a blunt-end duplex

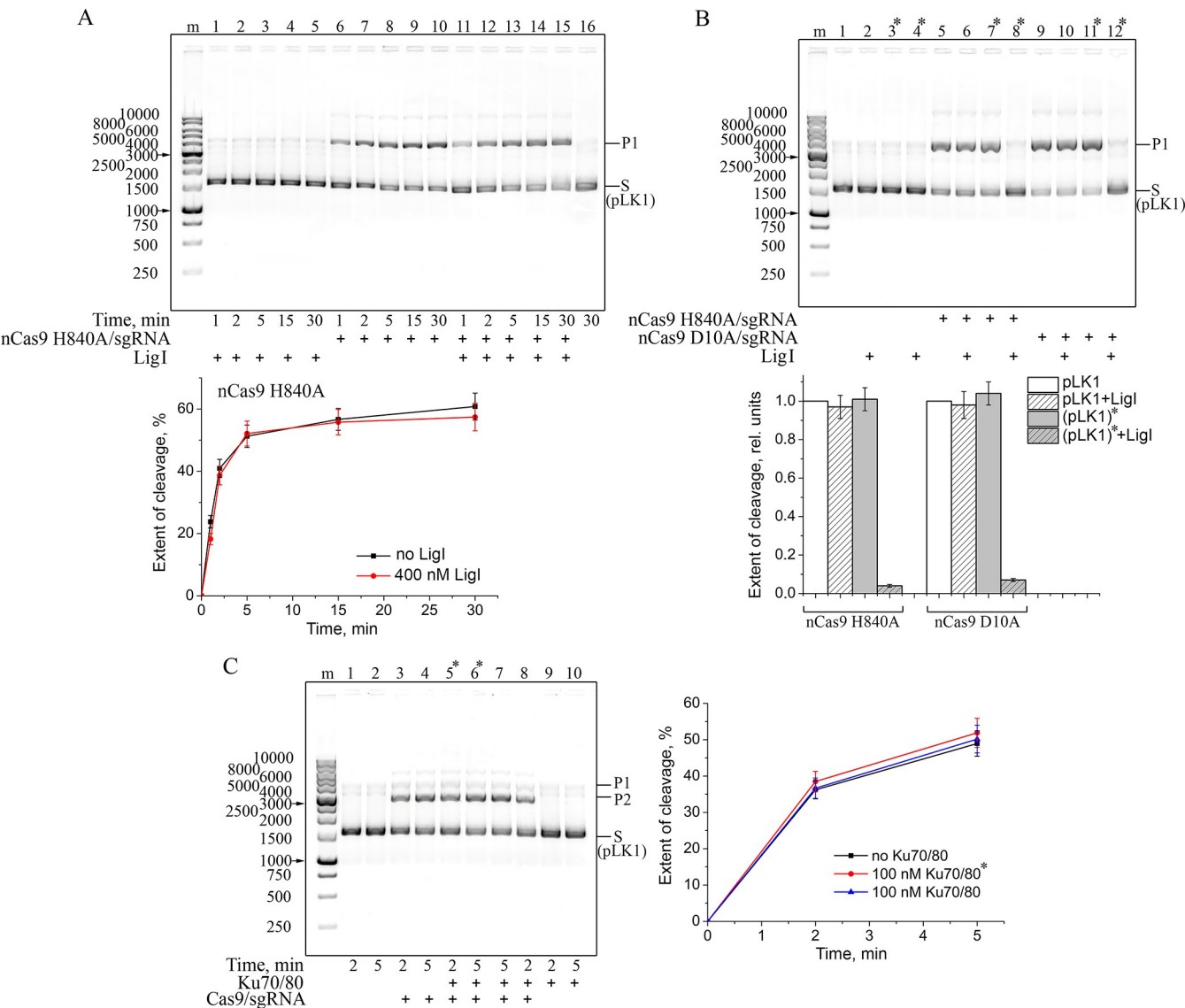

**Fig 6. Effects of SSB (DNA ligase I) and DSB (Ku70/80) repair sensors on the cleavage activity of nCas9 (Cas9).** (A) pLK1 (10 ng/μl) was incubated with nCas9 H840A/sgRNA (10 nM) and/or LigI (400 nM) as indicated. The curves show the time course of cleavage product (P1) appearance (the mean ± SD, $n = 3$). (B) pLK1 (10 ng/μl) was incubated with nCas9 D10A/sgRNA or nCas9 H840A/sgRNA (10 nM), and/or LigI (400 nM) for 20 min. The samples marked with an asterisk were heat-inactivated (70˚C for 5 min) before LigI addition and further incubated for 20 min. Bar charts show relative yields of the cleavage product under various reaction conditions (as specified in the legend) normalized to the respective value for each nickase without LigI and heating (the mean ± SD, $n = 3$). The sizes of DNA markers are indicated next to the gel images. (C) pLK1 (10 ng/μl) was incubated with Cas9/sgRNA (5 nM) and/or Ku70/80 (100 nM) as indicated. The samples marked with an asterisk were pre-incubated for 10 min with Ku70/80 before the Cas9/sgRNA addition. The sizes of DNA markers are indicated next to the gel image. The curves show the time course of cleavage product appearance (the mean ± SD, $n = 3$).

dsDNA1/2 (S9A Fig). Thus, Cas9 protects the generated DSB, shielding them from the Ku antigen.

## Cas9 is stimulated by RPA in a substrate-dependent manner

RPA is the major single-stranded DNA-binding protein in eukaryotic cells essential for replication, recombination, and repair. Its main function is the stabilization of single-stranded DNA regions in the unfolded state and their protection from endonucleases. RPA is a heterotrimer composed of 70-kDa, 32-kDa and 14-kDa subunits. *In vitro*, RPA can also bind

RNA and double-stranded DNA, but much less tightly than single-stranded DNA. Moreover, RPA is able to unwind the DNA duplex in an ATP-independent manner [29, 30]. Thus, RPA is a weak competitor for the interaction with double-stranded DNA, but it can potentially promote Cas9/sgRNA binding to DNA through destabilizing the duplex and facilitating sgRNA hybridization with the target DNA strand. We explored the influence of human RPA on the Cas9 activity using dsDNA1/2 as a substrate (Fig 7). Stimulation of the DNA cleavage was observed when the substrate was pre-incubated with RPA, alone or together with Cas9/sgRNA (Fig 7A and 7B). The Cas9 activity was not influenced by RPA when DNA was first pre-incubated with Cas9/sgRNA, most likely because Cas9/sgRNA has ~10-fold higher affinity than RPA for dsDNA1/2 (S4D and S9 Figs). No stimulation of Cas9 activity by RPA was observed when the pLK1 plasmid was used as a DNA substrate, even if RPA was pre-incubated with DNA (Fig 7C). This may be due to greater stability of the supercoiled plasmid DNA compared to the DNA duplex, which in turn makes DNA unwinding with RPA much more difficult.

To ensure that the lack of nick sensors' effect on Cas9 activity on covalently closed plasmid substrates is not dependent on the targeted sequence, we have repeated the experiments with another plasmid, pMSH2, carrying a fragment of human *MSH2* gene (S10 Fig). Although the overall efficiency of cleavage was different, most likely due to different protospacer GC content, we observed no inhibition or stimulation of Cas9 by PARP1, PARP2 (both either in the presence or in the absence of NAD$^+$), Ku70/80, or RPA. Thus, the ability of Cas9 to efficiently shield DSBs and SSBs from their sensors does not depend on the targeted DNA sequence.

## Competition of Cas9 with an RNA-interacting protein for sgRNA binding

YB1 is an intrinsically disordered 36-kDa DNA- and RNA-binding protein containing a conserved cold shock domain. YB1 participates in DNA repair and multiple mRNA-dependent processes, such as transcription, splicing, packaging, translation and regulation of mRNA stability [31]. YB1 is primarily located in the cytoplasm but its N-terminal part (residues 1–219) produced by the proteasomal cleavage translocates to the nucleus in response to environmental stresses such as genotoxic drugs, UV irradiation, oxidative stress, virus infection and hyperthermia [49]. To examine possible interplay between Cas9 and YB1 proteins, we explored the influence of the truncated human YB1 (1–219) on the Cas9/sgRNA-catalysed cleavage of dsDNA1/2 (Fig 8 and S11 Fig). No significant effect was detected even at a 50-fold molar excess of YB1 over Cas9 when YB1 and Cas9/sgRNA were either simultaneously added to the substrate or YB1 was pre-incubated with Cas9/sgRNA before the DNA addition (S11A Fig). The data suggest that YB1 can neither displace sgRNA from the preformed effector complex with Cas9, nor prevent formation of the ternary Cas9/sgRNA-DNA complex. No significant inhibition of the Cas9 activity was also detected when YB1 was pre-incubated with sgRNA with following addition of Cas9 and DNA (Fig 8A). We pursued this observation comparing the efficiency of DNA binding by Cas9, Cas9/sgRNA and YB1. The EMSA experiments were performed in the presence of Mg$^{2+}$ ions using dCas9 and dsDNA1/2. YB1 formed several complexes with different mobilities (S9C Fig). When both preformed Cas9/sgRNA and YB1 were present, the complex with the mobility corresponding to Cas9/sgRNA–DNA disappeared (at 15–20-fold excess of YB1 over Cas9/sgRNA); the complexes with the lowest mobility detected alongside with the YB1–DNA complexes apparently contained both YB1 and Cas9/sgRNA. Their formation was still detectable but much less efficient when Cas9 and sgRNA were added separately. At the highest YB1 concentration, Cas9/sgRNA–DNA–YB1 complexes became indistinguishable from the complexes containing only YB1. Most likely, the Cas9/sgRNA–DNA–YB1 complex is formed through the interactions of YB1 with the parts of sgRNA and DNA not covered by Cas9. The reported affinity of Cas9 for sgRNA (0.29 nM) [50] is at least

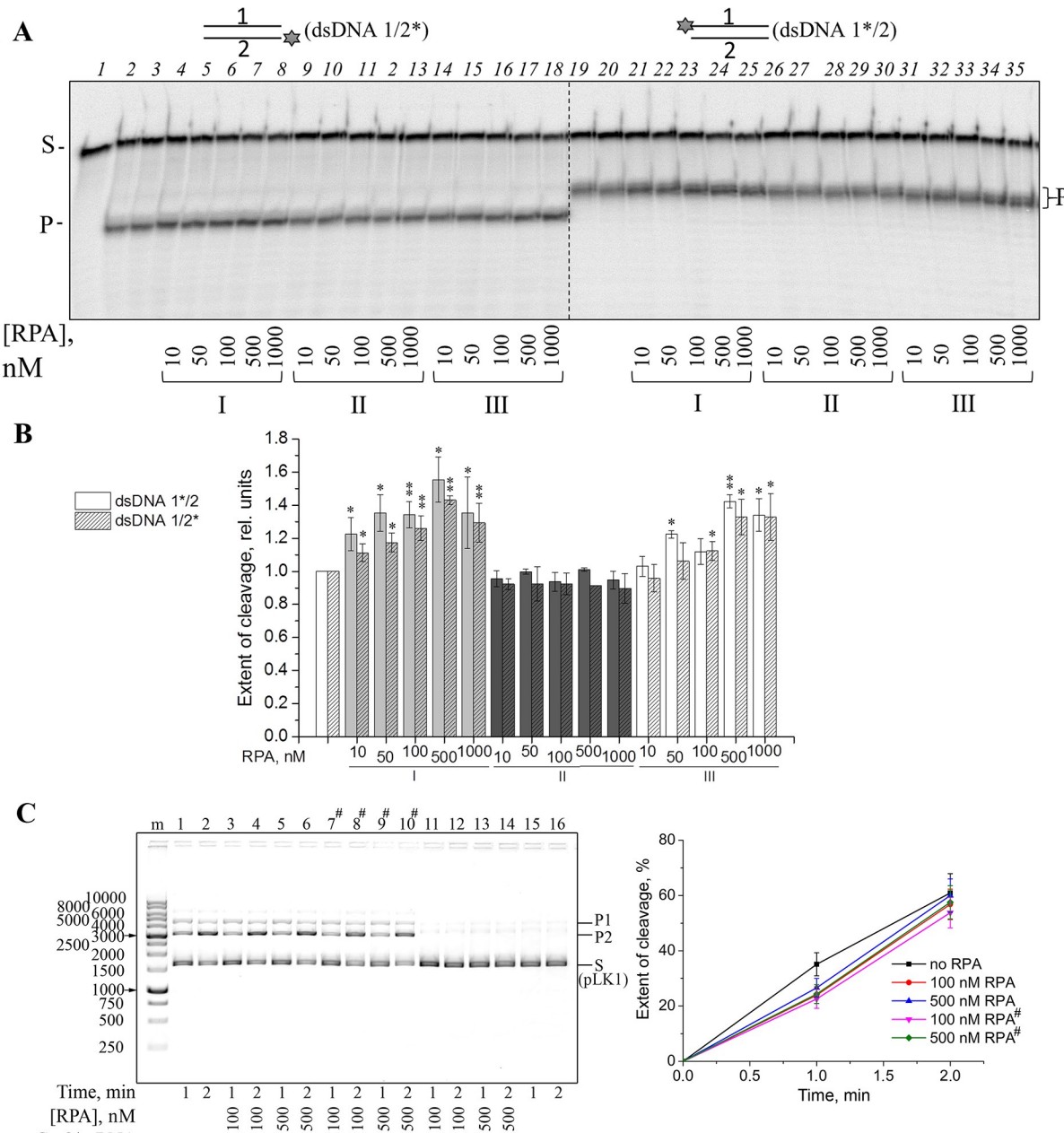

**Fig 7. Effects of RPA on the Cas9 activity.** (A) Cas9/sgRNA (10 nM) was incubated with dsDNA1*/2 or dsDNA1/2* (10 nM) and RPA (10–1000 nM) as indicated. The substrate was pre-incubated with RPA alone (I), Cas9/sgRNA alone (II) or RPA and Cas9/sgRNA (III) for 10 min. After addition of Cas9/sgRNA (I) or RPA (II) and 10 mM MgCl₂ the reaction was allowed to proceed for 30 min. (B) Relative extent of dsDNA1/2 cleavage under the specified reaction conditions normalized to the control without RPA. Values determined in the presence of RPA, which were statistically different from those in its absence, are marked p ≤ 0.05 (*), p ≤ 0.01 (**). (C) Cas9/sgRNA (10 nM) was incubated with pLK1 (10 ng/μl) and RPA as indicated. In the samples marked with a lattice, the plasmid was pre-incubated with RPA for 10 min before the addition of Cas9/sgRNA. The sizes of DNA markers are indicated next to the gel image. The plot shows the time course of cleavage product appearance (the mean ± SD, *n* = 3).

one order of magnitude higher than the affinity of YB1 for RNA (~10 nM) [51], indicating that sgRNA is the preferable ligand for Cas9. Together, our data demonstrate that YB1 does not interfere with DNA binding by Cas9/sgRNA but an effect of YB1 present at higher excessive amounts on the Cas9 activity cannot be excluded.

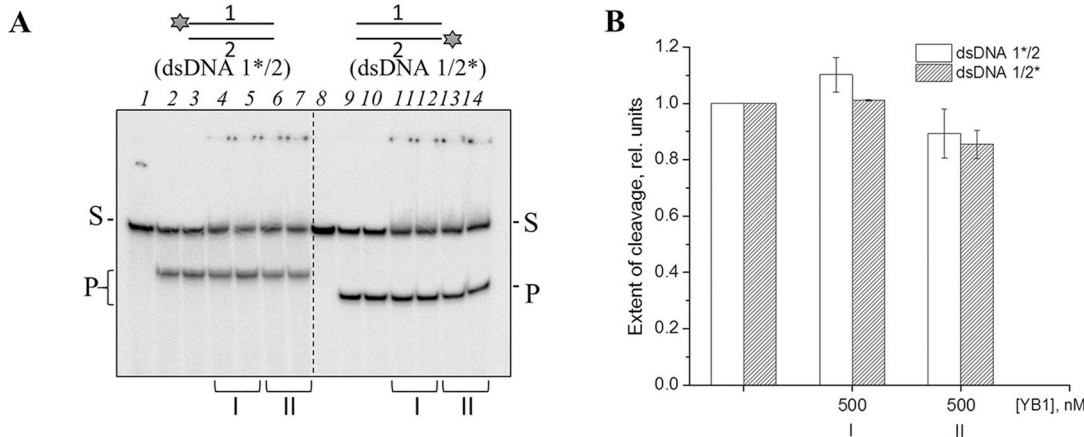

**Fig 8. Competition between Cas9 and YB1 for the interaction with sgRNA.** (A) Cas9 (40 nM) was incubated with sgRNA (20 nM), dsDNA1*/2 or dsDNA1/2* (10 nM) and YB1 as indicated. sgRNA was pre-incubated on ice for 40 min in the presence of $Mg^{2+}$ with either (I) Cas9 and YB1 or (II) YB1 alone. After the addition of dsDNA1/2 (I, II) and Cas9 (II) the reaction was allowed to proceed at 37˚C for 30 min. (B) Relative extent of dsDNA1/2 cleavage by Cas9 under the specified reaction conditions normalized to the control without YB1.

## Editing by Cas9 in HEK293 cells is independent of PARP1

Several approaches are widely used to measure the efficiency of Cas9-mediated editing in living cells: T7 endonuclease 1 digestion assay, Tracking of Indels by Decomposition (TIDE) assay, Next-Generation Sequencing (NGS), and Indel Detection by Amplicon Analysis (IDAA). Previously it was shown that TIDE estimates the editing efficiency as accurately as NGS and IDAA [52]. Therefore, we employed TIDE to assess the Cas9 editing efficiency in HEK293 and HEK293 *PARP1*$^{−/−}$ cells. Three protospacer sequences with different GC content (48–65%) were selected: H4, H6, and H9 in human *MAPT*, *LIPA* and *ABCA3* genes, respectively [52]. After transfection with pX458 plasmids encoding the sgRNAs, Cas9 nuclease and eGFP, the selected targets were amplified from the pool of transfected cells and subjected to Sanger sequencing. Since no homologous recombination template was provided, the editing reflects the repair of Cas9-mediated breaks by NHEJ. According to TIDE quantification, the combined editing efficiency at H4, H6, and H9 loci was 37.9%, 10.8%, and 3.5% in HEK293 and 36.5%, 9%, and 3.7% in HEK293 *PARP1*$^{−/−}$ cells (Fig 9). In all cases, +1 insertions were the predominant indels but other events together represented up to 50% of all mutations. Thus, PARP1 knockout does not change the efficiency of Cas9-mediated editing, at least by NHEJ, in cultured cells.

## Discussion

At present, genome editing in living cells mostly relies on the introduction of DSBs into DNA followed by repair through HR or NHEJ pathways. Although several base editing techniques have been developed to circumvent the need for DNA breakage, their outcome is usually less predictable and less efficient than in the break-and-repair procedure. However, the interaction of the RNA-targeted genome editors with the human cellular machinery responsible for DSB or SSB repair remains largely unexplored. For the most frequently used Cas9 nuclease, a large body of *in vitro* mechanistic data suggests that it releases the reaction product very slowly due to the persisting stable RNA/DNA heteroduplex [19, 20, 53, 54]. Cas9/sgRNA bound to its target buries ~11 nt of DNA at the PAM-proximal side of the scissile phosphodiester bond and ~19 nt at the PAM-distal side [55]. Hence, a DSB made by Cas9 could be shielded from cellular

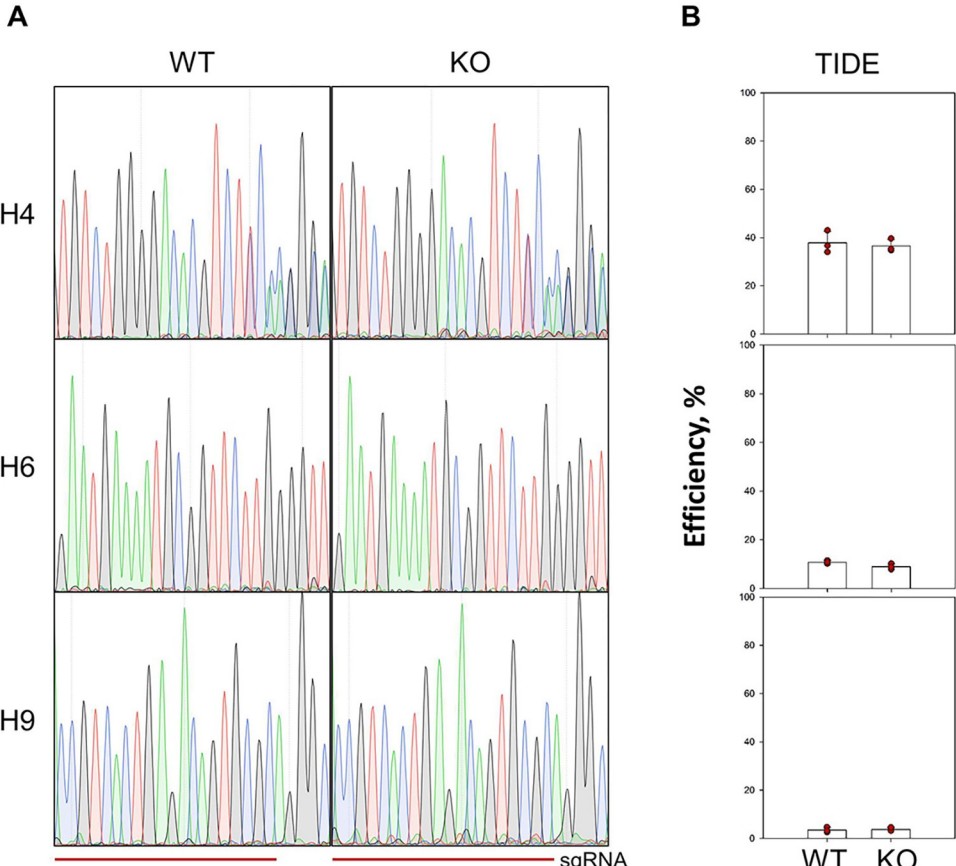

**Fig 9. Assessment of CRISPR/Cas9 efficiency in HEK293 and HEK293 *PARP1*⁻/⁻ cells by TIDE.** (A) H4, H6 and H9 loci targeted by sgRNA were amplified from HEK293 (WT) and HEK293 *PARP1*⁻/⁻ (KO) genomic DNA and sequenced. Representative sequence chromatograms are shown for each locus. The horizontal line below the sequence traces marks the sgRNA position. (B) Total efficiency of genome editing in the H4, H6, and H9 targets assessed by TIDE (*n* = 3; each dot represents a biological replicate). Sequencing chromatograms from all experiments are presented as S1 Appendix.

break sensors or, instead, the repair factors could favourably alter the dynamics of Cas9 action and efficiently compete for the DSB. As DSBs produced by Cas9 ultimately get repaired, they certainly become available to the basic HR/NHEJ machinery, but it is unclear whether the accompanying molecular events are the same as during the repair of DSBs of other origins. For example, DNA breaks made by Cas9 very strongly activate p53-dependent cell cycle arrest and cell death [56–58], which may suggest that the response to Cas9-induced damage is biased towards the cell elimination and against the repair. In this study, we have addressed the mechanism of possible interplay of Cas9 with several repair proteins involved in DSB and SSB repair, namely PARP1, PARP2, Ku70/80, LigI, RPA, and YB1, using established *in vitro* assays for the enzymatic activity and DNA, RNA or PAR binding by these proteins.

PARP1, the protein that was long regarded as a primary nick sensor, has now been established to play a role of the initial sensor of all DNA discontinuities including DSBs [14, 59–61]. PARP1 and its homolog PARP2 are multidomain proteins with a flexible architecture, capable of binding DNA in several modes [60, 62–67]. Once activated by DNA binding, PARPs catalyse the autoPARylation and PARylation of other targets, including DNA repair and chromatin proteins. These processes are essential for the DNA damage response (including chromatin

remodelling) and DNA repair pathways [9–11]. The autoPARylated PARP1 recruits the MRN complex, which is responsible for DSB repair initiation [68]. We revealed PARylation of Cas9 and its active form, the Cas9/sgRNA complex, catalysed by PARP2 in the presence of dsDNA substrate of Cas9. Wild-type PARP1 might also modify Cas9 to a degree but we were unable to separate the products of auto- and heteromodification, and PARP1 G972R short chain-producing mutant was apparently inactive in PARylation of Cas9. The results of other experiments (presented in Figs 1, 2 and 5) strongly suggest that PARPs and Cas9 interact with different binding sites in the DNA substrate, and that the DNA break site is shielded by Cas9 in both the substrate and product from binding and subsequent activation of PARPs. Moreover, the lack of PARPs effects on the Cas9 activity indicates that DNA binding and cleavage activities of Cas9 are not significantly modulated by PARylation of the enzyme. The *in vitro* mechanistic data were supported with cell experiments, in which the efficiency of editing in three different human genomic loci was also independent of PARP1 presence, further backing the idea of Cas9 orthogonality to PARP signalling.

Ku antigen, an abundant heterodimer of 70-kDa and 86-kDa subunits, is another major DSB sensor and a key player in c-NHEJ [69]. The binding of Ku protein to DNA ends is independent of the exact structure of the ends and is largely independent of DNA sequence [70]. Two Ku70/80 rings encircle dsDNA near both ends of a DSB and recruit DNA-PKcs, DNA ligase IV, XRCC4, XLF and APLF into a synaptic complex bringing together the DNA ends to be ligated [71–73]. The structural reasons of Ku's high affinity for DSB ends are presently unknown yet it is clear that Ku's preferable binding sites are sufficiently close to a break to fall within the Cas9/sgRNA footprint. No detectable impact of Ku70/80 on the Cas9 activity (Fig 6C) testifies to the inability of this DSB sensor to displace Cas9 from its product.

Mammalian DNA ligase, LigI, contributes mainly to base excision DNA repair (BER), being responsible for nick sealing at the final step, and is involved in other DNA repair pathways [48]. It is a likely candidate for modulating activity of Cas9 nickases due to competition for the interaction with SSB. We have shown that the products generated by nCas9 H840A/sgRNA or nCas9 D10A/sgRNA are ligated by LigI, but only after thermal inactivation of the nCas9 (Fig 6B). Thus, the two Cas9 mutants acting on the different DNA strands stably interact with the product, shielding it from accessibility for LigI.

RPA, a multifunctional protein composed of three subunits with several structurally related functional domains, is involved in various DNA metabolism pathways in eukaryotic cells [29, 30]. In addition to its primary ssDNA-binding activity, RPA can destabilize certain dsDNA structures. The destabilizing activity of RPA could play a role in promoting Cas9/sgRNA binding to DNA through facilitating sgRNA annealing to the target DNA strand. Indeed, the Cas9 cleavage activity on dsDNA was stimulated by the substrate pre-incubation with RPA, while the cleavage activity towards the plasmid substrate remained unaffected (Fig 7).

Another multifunctional protein involved in DNA- and RNA-dependent processes is YB1, mainly known as a regulator of gene expression [31]. Its N-terminal part (residues 1–219) produced by the proteasomal cleavage in response to DNA damage is localized in the nucleus [49]. YB1 is recognized as one of BER accessory factors [9, 28, 74], and its contribution to regulating PARP1 activity has been recently established [32]. To reveal an interplay between Cas9 and YB1, we explored the influence of the truncated YB1 (1–219) on the DNA binding and cleavage activities of Cas9/sgRNA (Fig 8 and S11 Fig). The results show the capability of YB1 to bind the productive Cas9/sgRNA-DNA complex without significant modulation of the Cas9 binding and cleavage activities.

Overall, our *in vitro* data suggest that Cas9-introduced DSBs and SSBs may not be easily drawn into the canonical break repair pathways in human cells. The breaks are effectively shielded from PARP1, Ku70/80, LigI and likely from other nick/break sensors and processing

enzymes until their release by Cas9, which may take a long time unless some additional mechanisms ensuring Cas9 displacement exist. Notably, a question yet to be resolved is whether any additional intermediates arise after the DNA has been cut. High-speed atomic force microscopy suggests that *in vitro* wild-type Cas9 slowly releases the PAM-proximal DNA segment, which forms only three base pairs with sgRNA, while remaining tightly bound to the PAM-distal segment through the remaining 17 bp heteroduplex [53]. If this mechanism also operates in the cell, the product will essentially behave as a DSB with one break end free and another held by a large protein globule, akin to stalled topoisomerase adducts [75]. Such cross-links are repaired by proteolytic degradation followed by end cleaning by tyrosyl-DNA phosphodiesterases [76], and while the latter step is not required for removal of non-covalently bound proteins, it would be interesting to see whether Cas9 interacts with the proteases participating in protein–DNA cross-links repair [77]. Also, the post-incision activity of the RuvC domain of Cas9 generates a highly heterogeneous population of recessed ends in both $3'{\rightarrow}5'$ and $5'{\rightarrow}3'$ directions in the non-target strand [78], providing ample possibilities for entry of other exonucleases and microhomology-mediated annealing after the product release.

Aside from the events at the breaks, expression of Cas9 and sgRNA (with no homology in the human genome) in HeLa or HEK293T cells has been reported to have minimal impact on the transcriptome, proteome, protein synthesis or phosphorylation, and histone modifications, suggesting near orthogonality, at least in human cells in culture [79]. In the same study, the interactome of Cas9 was shown to consist predominantly of highly abundant cellular proteins that most likely either interact with Cas9 non-specifically or are coincidentally trapped during the affinity purification (CRAPome) [79]. A recent study of Cas9 interactions with human RNA identified a subset of the human transcriptome that Cas9 haphazardly binds but found no apparent consequences for the cell beyond partial sequestration of these transcripts [80]. Our data thus fit well into the view of Cas9 as an enzyme mostly orthogonal to the human signalling pathways.

## Methods

### Proteins

All proteins used in this study were purified essentially as described [6, 32, 46, 81–84]. *Streptococcus pyogenes* Cas9 and its mutant forms nCas9 D10A, nCas9 H840A (nickases) and dCas9 (catalytically inactive Cas9 harbouring the double D10A/H840A mutation) were overproduced and purified from *E. coli* BL21(DE3) using the expression plasmids pMJ806, pMJ825, pMJ826 and pMJ841 (a gift from Dr. Jennifer Doudna; Addgene plasmids #39312, http://n2t.net/addgene:39312, RRID:Addgene_39312; #39315, http://n2t.net/addgene:39315, RRID:Addgene_39315; #39316, http://n2t.net/addgene:39316, RRID:Addgene_39316; #39318, http://n2t.net/addgene:39318; RRID:Addgene_39318). The purified Cas9 proteins were dialyzed against the storage buffer containing 50 mM Tris–HCl (pH 8.0), 200 mM NaCl, 5 mM DTT and 40% glycerol and stored at −30˚C. Human PARP1 and murine PARP2 were produced in Sf9 insect cells using the expression plasmids kindly provided by Dr. Valérie Schreiber (University of Strasbourg, France). The human PARP1 G972R mutant was expressed in *E. coli* BL21(DE3) using the pET-32a-based expression plasmid [46]. Human RPA was expressed in *E. coli* BL21(DE3) using the expression plasmid kindly provided by Dr. Marc S. Wold (University of Iowa, USA). Human LigI was produced in *E. coli* BL21(DE3) using the expression plasmid kindly provided by Dr. Robert A. Bambara (University of Rochester Medical Center, USA). Ku antigen was purified from HeLa cell extract and was kindly provided by Dr. Svetlana N. Khodyreva (SB RAS Institute of Chemical Biology and Fundamental Medicine, Novosibirsk, Russia). A truncated form of human YB1 containing the N-terminal residues 1–219 was

produced in *E. coli* BL21(DE3) using the pET-32a-based expression plasmid [32]. Homogeneity of the purified proteins was verified by SDS-PAG electrophoresis. The enzymatic activities of Cas9, PARP1, PARP2 and LigI were verified as described previously [6, 35, 82, 85]. Phage T4 DNA ligase, Taq DNA polymerase, T7 RNA polymerase, T4 polynucleotide kinase, proteinase K and restriction endonucleases were from various commercial sources.

## Cell lines and whole-cell extracts

The HEK293 cell line was obtained from Thermo Fisher Scientific, Waltham, MA, USA, #R70507. The HEK293 *PARP1*$^{-/-}$ knockout cell line used in the work was constructed and characterized earlier [86]. The residual PARP1 expression level was 10% of the wild type. Whole-cell extracts were prepared from the parent HEK293 and HEK293 *PARP1*$^{-/-}$ cells as described [87] and stored in aliquots at −70˚C. Protein concentration in the extracts was determined with the Bradford assay using BSA as a standard.

## sgRNA for *in vitro* experiments

Single-guide RNA (sgRNA; 5′−pppGGAUAACUCAAUUUGUAAAAAAGUUUUAGAGCUAGAAA UAGCAAGUUAAAAUAAGGCUAGUCCGUUAUCAACUUGAAAAAGUGGCACCGAGUCGGUGCUU UU−3′) was synthesized by *in vitro* transcription with T7 RNA polymerase and purified by electrophoresis in 8% polyacrylamide gel containing 7.2 M urea as described [88].

## DNA substrates

All oligonucleotides were synthesized in the Laboratory of Medicinal Chemistry (SB RAS Institute of Chemical Biology and Fundamental Medicine, Novosibirsk, Russia). Two types of DNA substrates were used to follow the Cas9-catalysed reaction. The 35-mer DNA duplex (dsDNA1/2) was prepared by annealing the non-target strand (DNA1) with the target strand (DNA2) (S1 Table). If necessary, either DNA1 or DNA2 were $^{32}$P-labelled using [γ-$^{32}$P]ATP (Laboratory of Biotechnology, SB RAS Institute of Chemical Biology and Fundamental Medicine, Novosibirsk, Russia) and phage T4 polynucleotide kinase. The plasmid substrate pLK1 was constructed by cloning a DNA fragment containing the protospacer and PAM sequences into pBlueScript II SK(−) vector at the *Xho*I–*Eco*RI restriction sites (S1 Fig).

## Synthesis and purification of bulk poly(ADP-ribose)

$^{32}$P-labelled PAR was synthesized enzymatically from [$^{32}$P]NAD$^+$ and purified (without fractionation) as described previously [42].

## Cas9 activity assay

Cas9/sgRNA complex was preassembled at 4˚C by mixing Cas9 and sgRNA at an equimolar ratio and incubating for 15 min. The activity of the complex was assayed in reaction mixtures containing 50 mM Tris–HCl (pH 7.5), 100 mM NaCl, 10 mM MgCl$_2$, 1 mM DTT, 0.1 mg/ml BSA, and either 10 nM $^{32}$P-labelled dsDNA1/2 or 10 ng/μl pLK1. When indicated the reaction mixtures were supplemented with 10–1000 nM PARP1 or PARP2 (with or without 50–500 μM NAD$^+$), 10–1000 nM XRCC1, 10–1000 nM RPA, 100–400 nM LigI, 100 nM Ku, 10–1000 nM YB1 or 1–4 μg of cell extract (HEK293 or HEK293 *PARP1*$^{-/-}$). The mixtures were incubated with 10 nM Cas9/sgRNA (unless otherwise stated) at 37˚C for 30 min. The reactions with DNA1/2 were terminated by adding 0.25 vol of denaturing solution consisting of 95% formamide, 50 mM EDTA (pH 8.0), 0.01% bromophenol blue and 0.01% xylene cyanol and heating for 5 min at 95˚C. In the experiments involving YB1, the reactions were stopped by

adding an equal volume of denaturing solution consisting of 8 M urea, 20 mM EDTA (pH 8.0), 0.01% bromophenol blue and 0.01% xylene cyanol. The reaction products were separated by electrophoresis in a 20% denaturing polyacrylamide gel. The mixtures containing the pLK1 plasmid DNA substrate were incubated with 0.5–10 nM Cas9/sgRNA at 37˚C for time period indicated in each experiment; the reactions were stopped with addition of a stop solution (0.25 vol) containing 100 mM EDTA (pH 8.0), 1.2% SDS, 30% glycerol, and 0.01% bromophenol blue. The reaction products were separated by electrophoresis in 1% GelRed dye stained agarose gel in 1×TAE buffer. All gels were visualized on a Typhoon FLA 9500 imaging system (GE Healthcare, Chicago, IL) and quantified using Quantity One software (Bio-Rad Laboratories, Hercules, CA). The extent of plasmid DNA cleavage ($\Theta$) was calculated using the equations:

$$\text{SSB}, \% = \frac{I_r}{I_r + I_l + I_{sc}} \cdot 100\%$$

$$\text{DSB}, \% = \frac{I_l}{I_r + I_l + I_{sc}} \cdot 100\%$$

where SSB is the amount of breaks in one strand of the plasmid DNA, DSB is the amount of breaks in both strands of plasmid DNA, $I$ is the fluorescence intensity of bands corresponding to the relaxed ($I_r$), linear ($I_l$) and supercoiled ($I_{sc}$) forms of the plasmid. The yield of DSB/SSB generated in the enzymatic reaction (Cas9/nCas9-catalysed) was calculated by subtraction of the amount of breaks present in the substrate (without Cas9) from the total amount of DSB and SSB detected in the presence of Cas9 (with Cas9):

$$DSB_{(Cas9\ catalyzed)}, \% = DSB_{(with\ Cas9)} - DSB_{(w/o\ Cas9)}$$

$$SSB_{(Cas9\ catalyzed)}, \% = SSB_{(with\ Cas9)} - SSB_{(w/o\ Cas9)} \cdot \left( 1 - \frac{DSB_{(Cas9\ catalyzed)}}{100\%} \right)$$

$$\Theta, \% = DSB_{(Cas9\ catalyzed)} + SSB_{(Cas9\ catalyzed)}$$

The extent of the plasmid DNA substrate self-cleavage with generation of products co-migrating with the Cas9-induced cleavage products did not exceed 6% in the initial substrate and 10% after a 30-min incubation. All the experiments were performed at least three times. The data were analysed for statistically significant differences by Student's $t$-test.

## Electrophoretic mobility shift assay

The affinity of Cas9 and its mutants for PAR or DNA was measured by electrophoretic mobility shift assay (EMSA). The protein (0.1–4 μM in experiments with PAR, 0.002–0.064 μM in experiments with dsDNA) was incubated with [$^{32}$P]PAR (~10nM) or [$^{32}$P]dsDNA1/2 (10 nM) in a 10 μl mixture containing 50 mM Tris–HCl (pH 7.5), 100 mM NaCl, 1 mM DTT, 0.1 mg/ml BSA (only in experiments with dsDNA) and 10 mM MgCl$_2$ (if indicated) at 4˚C for 30 min. After addition of glycerol and bromophenol blue (to the final concentrations of 10% and 0.1% respectively), the samples were subjected to electrophoresis at 4˚C in a 5% non-denaturing polyacrylamide gel in 0.5×TBE buffer (pH 8.0). The gels were visualized and quantified as above. The data were fitted to a Hill equation:

$$\Theta = \frac{\Theta_\infty}{1 + \left( \frac{\text{EC}_{50}}{\text{C}} \right)^n}$$

where Θ is the fraction of protein-bound ligand (calculated as amount of the complex divided by the total ligand amount) at a given concentration (C) of the protein, $\Theta_\infty$ is the maximal extent of ligand binding (i.e. the fraction of the ligand bound at the saturating protein concentration), $EC_{50}$ is the protein concentration at which $\Theta = \Theta_\infty/2$, and $n$ is the Hill coefficient, which determines the slope of the nonlinear curve. Experiments with PARP1/PARP2, RPA, Ku70/80, and YB1 binding to dsDNA1/2 were performed under the same conditions.

To analyse dCas9/sgRNA, PARP1 and PARP2 binding to pLK1 plasmid DNA, the protein (5–500 nM concentrations) was incubated with 10 ng/μl pLK1 in a 8 μl mixture containing 50 mM Tris–HCl (pH 7.5), 100 mM NaCl, 10 mM $MgCl_2$ and 1 mM DTT at 4°C for 30 min. After addition of glycerol and bromophenol blue (to the final concentrations of 10% and 0.1% respectively), the samples were subjected to electrophoresis at 4°C in a 1% GelRed dye stained agarose gel. To study protein competition for plasmid DNA binding, the reaction mixture contained PARP1 at fixed concentration (200 nM) and Cas9 (Cas9/sgRNA) at 100–500 nM.

### AutoPARylation of PARP1 or PARP2 and PARylation of Cas9

The reaction mixture contained 50 mM Tris–HCl (pH 7.5), 100 mM NaCl, 10 mM $MgCl_2$, 1 mM DTT, 0.8–8 μM [$^{32}$P]NAD$^+$, 1 μM dsDNA1/2 or 100 ng/μl pLK1 (supercoiled or *Eco*RI-linearized), 200–600 nM PARP1 or PARP1 G972R and 0.1–4 μM Cas9 (supplemented or not with sgRNA). In some cases, before initiation of ADP-ribosylation, DNA was pre-incubated with Cas9/sgRNA for 20 min at 37°C, then heated for 5 min at 70°C for enzyme inactivation and centrifuged for 10 min at 9,500 rpm. The reaction was initiated by addition of [$^{32}$P]NAD$^+$. After incubating the mixtures at 37°C for 20–60 min, the reactions were terminated by the addition of SDS-PAGE sample buffer and heating for 2 min at 90°C. The reaction products were analysed by 10% SDS-PAGE with subsequent phosphor imaging on a Typhoon FLA 9500 system. All the experiments were performed at least three times. The data were analysed for statistically significant differences by Student's *t*-test.

### Genome editing efficiency assessment in PARP1-deficient cells

The protospacer sequences of H4, H6 and H9 (S1 Table) human genomic targets [47] were cloned at the *Bbs*I site in pSpCas9(BB)-2A-GFP (pX458) plasmid [89] (a gift from Dr. Feng Zhang; Addgene plasmid #48138; http://n2t.net/addgene:48138; RRID: Addgene_48138). To assess the editing efficiency, 5×10$^5$ HEK293 or HEK293 *PARP1$^{-/-}$* cells were seeded per well of a six-well plate and transfected with 2.5 μg of the plasmid 24 h later using Lipofectamine 3000 (Thermo Fisher Scientific, Waltham, MA). Twenty-four hours post-transfection, the cells were sorted on Bio-Rad S3e Cell Sorter. To obtain genomic DNA, 3×10$^4$ EGFP-positive cells were collected and lysed in the buffer containing 10 mM Tris–HCl (pH 8.0), 10 mM EDTA, 3.5% Igepal CA630, 3.5% Tween 20, and 0.4 mg/ml proteinase K for 3 h at 65°C. The target region was PCR-amplified using Taq DNA polymerase (S1 and S2 Tables) and analysed by Sanger sequencing. The efficiency of indel formation was calculated using TIDE [90].

### Supporting information

**S1 Table. Oligonucleotides used in this study.**
(DOCX)

**S2 Table. PCR protocol for genomic targets amplification.**
(DOCX)

**S1 Fig. Substrates used in the Cas9 activity assays.**
(DOCX)

**S2 Fig. Cleavage of the oligonucleotide substrate by Cas9 nickase mutants in the presence of PARP1 and PARP2.**
(DOCX)

**S3 Fig. Cleavage of the plasmid substrate by Cas9 nickase mutants in the presence of PARP1 and PARP2.**
(DOCX)

**S4 Fig. Binding of Cas9 and its mutant forms to the dsDNA substrate of Cas9.**
(DOCX)

**S5 Fig. Binding of PARP1 and PARP2 to the dsDNA substrate of Cas9.**
(DOCX)

**S6 Fig. Binding of dCas9/sgRNA complex, PARP1 and PARP2 to plasmid DNA.**
(DOCX)

**S7 Fig. Protection of Cas9-generated dsDNA1/2 cleavage product from degradation in cell extracts.**
(DOCX)

**S8 Fig. Effects of DNA ligase I on the nickase activity of Cas9.**
(DOCX)

**S9 Fig. Binding of Ku70/80, RPA and YB1 to the dsDNA substrate of Cas9.**
(DOCX)

**S10 Fig. Cleavage of the pMSH2 plasmid substrate by Cas9/sgRNApMSH2 in the presence of different proteins.**
(DOCX)

**S11 Fig. Effects of YB1 on the cleavage activity of Cas9.**
(DOCX)

**S1 Appendix. Sequencing chromatograms from all cell transfection experiments shown in Fig 9.**
(ZIP)

**S1 Raw images. Original gel images.**
(PDF)

## Acknowledgments

We are thankful to Dr. Svetlana N. Khodyreva for providing us with purified Ku antigen protein and Konstantin N. Naumenko for providing us with purified YB1. DNA sequencing was performed at the SB RAS Genomics Core Facility.

## Author Contributions

**Conceptualization:** Ekaterina A. Maltseva, Inna A. Vasil'eva, Nina A. Moor, Dmitry O. Zharkov, Olga I. Lavrik.

**Formal analysis:** Ekaterina A. Maltseva, Inna A. Vasil'eva, Daria V. Kim.

**Funding acquisition:** Dmitry O. Zharkov, Olga I. Lavrik.

**Investigation:** Ekaterina A. Maltseva, Inna A. Vasil'eva, Daria V. Kim.

**Methodology:** Ekaterina A. Maltseva, Inna A. Vasil'eva, Daria V. Kim.

**Project administration:** Dmitry O. Zharkov, Olga I. Lavrik.

**Resources:** Ekaterina A. Maltseva, Inna A. Vasil'eva, Daria V. Kim, Nadezhda S. Dyrkheeva, Mikhail M. Kutuzov, Ivan P. Vokhtantsev, Lilya M. Kulishova.

**Supervision:** Nina A. Moor, Dmitry O. Zharkov.

**Validation:** Ekaterina A. Maltseva, Inna A. Vasil'eva, Daria V. Kim.

**Visualization:** Ekaterina A. Maltseva, Inna A. Vasil'eva, Daria V. Kim.

**Writing – original draft:** Ekaterina A. Maltseva, Inna A. Vasil'eva, Daria V. Kim, Dmitry O. Zharkov.

**Writing – review & editing:** Ekaterina A. Maltseva, Inna A. Vasil'eva, Nina A. Moor, Dmitry O. Zharkov, Olga I. Lavrik.

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
