## [Decision Letter · Decision Letter 0]

18 Jul 2023

PONE-D-23-20117Cas9 is mostly orthogonal to human systems of DNA break sensing and repairPLOS ONE

Dear Dr. Lavrik,

Thank you for submitting your manuscript to PLOS ONE. After careful consideration, we feel that it has merit but does not fully meet PLOS ONE’s publication criteria as it currently stands. Therefore, we invite you to submit a revised version of the manuscript that addresses the points raised during the review process.

We look forward to receiving your revised manuscript.

Kind regards,

Zhiming Li, Ph.D.

Academic Editor

PLOS ONE

Reviewers' comments:

Reviewer's Responses to Questions

**Comments to the Author**

1. Is the manuscript technically sound, and do the data support the conclusions?

Reviewer #1: Partly

Reviewer #2: Partly

2. Has the statistical analysis been performed appropriately and rigorously? 

Reviewer #1: No

Reviewer #2: I Don't Know

3. Have the authors made all data underlying the findings in their manuscript fully available?

Reviewer #1: No

Reviewer #2: Yes

4. Is the manuscript presented in an intelligible fashion and written in standard English?

Reviewer #1: Yes

Reviewer #2: Yes

5. Review Comments to the Author

Reviewer #1: In this study, Maltseva and colleagues used standard biochemistry techniques to investigate whether mammalian single-strand break and double-strand break signalling and catalytic proteins engage with or affect activity of Streptococcus pyogenes Cas9. They report that PARP1, PARP2, Ku antigen, RPA, LIG1 nor YB1 significantly impact Cas9/sgRNA binding to the DNA target or cleavage activity. The group demonstrates PARylation of Cas9 by PARP2, but this did not appear to significantly affect binding of Cas9 to the target DNA or cleavage activity unless in high concentrations. Thus, they conclude that Cas9 DSB-formation follows the canonical DSB signaling pathway in which Cas9-induced breaks are then recognized and repaired.

While this study is well written, thorough in the biochemical assays, and for the first time to my knowledge investigates whether the mammalian DSB repair mechanisms impact DSB formation, several concerns should be addressed.

Major concerns

1. There is no statistics reported for any of the experiments. The authors make claims throughout the entire manuscript of “no significant change” (i.e., line 96) or “significant change” (line 21), yet there are no statistics described in the methods, results, or figures. Nearly all assays have at least three data points and even the “simplest” statistics (i.e., Student’s T test) need to be applied to all data sets for any conclusion statement to be made.

2. Clarity in some figures need to be improved. There are a lot of difference conditions and samples used throughout this work. This strengthens the scientific rigor of the work, but it is difficult to follow in some cases. Specific examples include:

•Fig. 2., label the substrate (S) and product (P) to the right of the gel images, as done with the dsDNA experiments.

•Fig. 8, what does “START” on the gel image mean? This should be defined in the figure legend.

3. Some overstatement of claims. The authors claim that Cas9 DSB formation is orthologous to human DSB formation and signalling, although some of the proteins used in their in vitro assays are not human (i.e., PARP2 is murine), or species is not stated (i.e., RPA, Ku antigen, etc). This needs to be clarified in the results section of the paper, rather than hidden in the methods and conclusions should be more conservative.

4. Additional experiments to strengthen the results of the paper. To my knowledge, there are no reports that suggest how Cas9 is ultimately removed from the DNA target after cleavage. The authors describe in the Discussion the high affinity for the DNA target and slow removal of Cas9. Is it possible that Cas9 is removed in an MRN-dependent manner, as Spo11 is removed from DSBs during meiosis in yeast and humans? This experiment could strengthen the results of this work as it could provide a mechanism in which Cas9 is ultimately removed.

Minor concerns:

1. The paper is well-written, but there are a few grammatical errors, throughout, including:

• Line 70, “DSB” should be “a DSB”

• Line 95 “taken” should be “used”

• Line 99 “we have then” should be “we then”

• Line 276 “SSB” should be “an SSB”

• Line 300, “except for the S and G2/M phases of the cell cycle” should be “except during the S and G2/M phases of the cell cycle”

• Line 471-472 “SSBs may be not easily drawn into” should be “SSBs may not be easily drawn into”

2. Clarifying rationale/approach in a few experiments, including:

• Fig. 1, why is the plasmid incubated for only 2 min, but the dsDNA substrate for up to 30 min? Why are there multiple time points for the plasmid assays in panels C and D? It was helpful in Line 96 to provide the basal level of cleavage of the dsDNA substrate. Could this also be provided for the plasmid assay? For Fig 1 legend, please clarify by labeling panel C and D that the pLK1 plasmid was used in either the figure itself or the figure legend.

• Define dCas9 earlier in the results section. The first time it is introduced in Fig. S4, the “dCas9” should be defined as the double mutant. I had to look for this in the methods to determine what it meant.

• In Figure 2, line 168, the authors describe the mobility shift in the pLK1 plasmid as non-specific binding to the plasmid. It would be helpful to demonstrate this by repeating the experiment with an pLK1 plasmid without the dsDNA target sequence.

• Figure 4, what do the numbers on the X axis of both the bar graphs mean? Do they correspond to the lane numbers on the respective gels? If yes, please add this information in the figure legend. If no, please remove, as they are a bit confusing.

• Figure 5, similar to Fig. 4, what do the numbers on the x axis of the bar graph mean? I don’t think the correspond to lane numbers of the corresponding gel, so they should probably be removed.

• For the LIGI experiment, I’m a bit confused about the rationale for the experiment. If LIGI engages with a broken DNA molecule, why would it compete with Cas9 at a SSB? Wouldn’t Cas9 cleavage need to be completed before LIGI would bind? Or is the rationale that Cas9 may create a SSB that competes with LIGI and prevents a complete cleavage (DSB) reaction? If yes, clarifying this may strengthen the proposed rationale for the experiment.

3. Other minor concerns:

• Line 165, the authors mention the impact at 100 mM, but also a greater shift at 500 nM of PARP1 is apparent

• For the TIDE experiment, I do not believe this detects HR repair since an HDR substrate was not transfected. This should be mentioned in the Discussion or interpretation of the results.

• Line 622, how much of each vector was transfected and how many cells were transfected? This is particularly important if the transfection efficiency impacts DSB formation (and subsequent repair).

• Fig. 9 includes three biological replicates, but the panel A only provides one sequencing read. For data sharing purposes, the other sequences should be provided, or in the very least, the figure legend should state that these are representative chromatograms from three biological replicates.

Reviewer #2: This manuscript by Maltseva and colleagues investigates how different factors involved in DNA repair or other DNA/RNA transaction mechanisms could potentially influence Cas9 behavior, both in terms of DNA binding and cleavage activity. The manuscript is clear and the results that are presented appear overall of good quality. Nevertheless, the relevance of some of the findings is questionable. This is particularly the case of the results regarding Cas9 ability to bind PAR as well as PARP2-dependent APD-ribosylation of Cas9, which appear somehow out of context and do not bring much to the current study. I would then suggest to remove these data from the manuscript. Regarding the other findings, I advise the authors to address the following concerns in order to consolidate their main conclusions.

1) The different results obtained in this manuscript rely on a single target DNA sequence and sgRNA. It would be interesting to control that, at least for a set of key experiments, similar results can be obtained with a different target sequence and sgRNA.

2) In several instances such as with PARP1 and Ku, the authors aim to look at a possible competition between Cas9 and the repair factors for binding at the DSB or nick generated by the Cas9. However, with the dsDNA, the results are most probably blurred by the ability of PARP1 and Ku to bind to the ends of the oligo. The use of the pLK1 plasmid is not very helpful here since the large size of this plasmid does not directly allows to assess protein binding by gel shift. It would be important to investigate further a direct competition between Cas9 and PARP1 or Ku at the Cas9-generated nick or DSB by using for example dumbbell DNA oligos that would prevent PARP1 or Ku binding at the DNA ends and also be small enough to be able to assess protein binding by gel shift assay.

3) On the gels shown on figure S4, the authors should further comment on the existence of two separated bands in the DNA bound fraction.

4) On Fig 5, the author investigate how Cas9 mediated DSB on the pLK1 plasmid can induce PARP1 ADP-ribosylation. Since SSB, rather than DSB, is the primary substrate of PARP1, it would be interesting to perform the same experiment using the nickase version of the Cas9.

5) On Fig 7C, the authors show that RPA is unable to stimulate Cas9 activity, in contrast to what they observed with the DNA duplex. The authors should comment further this difference between the two assays.

6) Information regarding sample size and other relevant statistics are generally missing and should be included.

6. PLOS authors have the option to publish the peer review history of their article (what does this mean?). If published, this will include your full peer review and any attached files.

Reviewer #1: No

Reviewer #2: No

---

## [Author Response · Author response to Decision Letter 0]

3 Aug 2023

Dear Reviewers,

Thank you for critical reading and suggestions. Our detailed responses are listed below.

Reviewer #1: In this study, Maltseva and colleagues used standard biochemistry techniques to investigate whether mammalian single-strand break and double-strand break signalling and catalytic proteins engage with or affect activity of Streptococcus pyogenes Cas9. They report that PARP1, PARP2, Ku antigen, RPA, LIG1 nor YB1 significantly impact Cas9/sgRNA binding to the DNA target or cleavage activity. The group demonstrates PARylation of Cas9 by PARP2, but this did not appear to significantly affect binding of Cas9 to the target DNA or cleavage activity unless in high concentrations. Thus, they conclude that Cas9 DSB-formation follows the canonical DSB signaling pathway in which Cas9-induced breaks are then recognized and repaired.

While this study is well written, thorough in the biochemical assays, and for the first time to my knowledge investigates whether the mammalian DSB repair mechanisms impact DSB formation, several concerns should be addressed.

Major concerns:

1. There is no statistics reported for any of the experiments. The authors make claims throughout the entire manuscript of “no significant change” (i.e., line 96) or “significant change” (line 21), yet there are no statistics described in the methods, results, or figures. Nearly all assays have at least three data points and even the “simplest” statistics (i.e., Student’s T test) need to be applied to all data sets for any conclusion statement to be made.

All the experimental data were analysed by Student’s t-test to reveal statistically significant differences. The following information has been added to the Methods section (p. 25, lines 605-606; p. 26, lines 640-642): “All the experiments were performed at least three times. The data were analysed for statistically significant differences by Student’s t-test”. Statistically significant differences are now indicated in the revised Figures 4, 5 and 7.

2. Clarity in some figures need to be improved. There are a lot of difference conditions and samples used throughout this work. This strengthens the scientific rigor of the work, but it is difficult to follow in some cases. Specific examples include:

•Fig. 2., label the substrate (S) and product (P) to the right of the gel images, as done with the dsDNA experiments.

Fig. 2 represents binding data obtained from EMSA experiments. The positions of free DNA/sgRNA and their complexes with the individual proteins (Cas9, dCas9, PARP2 and PARP1) or the Cas9/dCas9 mixture with PARP1 in the gels are indicated in the revised Figure 2. Supporting Figure 6 has been edited in the same way.

•Fig. 8, what does “START” on the gel image mean? This should be defined in the figure legend.

“Start” indicated the position of the gel wells. This legend has now been removed from the revised Figure 8.

3. Some overstatement of claims. The authors claim that Cas9 DSB formation is orthologous to human DSB formation and signalling, although some of the proteins used in their in vitro assays are not human (i.e., PARP2 is murine), or species is not stated (i.e., RPA, Ku antigen, etc). This needs to be clarified in the results section of the paper, rather than hidden in the methods and conclusions should be more conservative.

We indeed used murine PARP2, while all the other proteins were human. This information has been added to the Results section for each protein at the first mention (p. 4, lines 93-94; p. 12, line 288; p. 13, line 316; p. 14, line 331; p. 15, line 362). It should be noted that murine PARP2 is highly homologous to its human counterpart as indicated by 87% identity between their catalytic domains and 62% identity between the N-terminal domains [Amé et al., J. Biol. Chem. 1999;274:17860-17868]. We have added this information and the respective reference to the revised text: the sentence “We have investigated whether PARP1 or PARP2 affect the activity of Cas9/sgRNA in the absence and in the presence of NAD+” has been replaced with “We have investigated whether human PARP1 or murine PARP2 affect the activity of Cas9/sgRNA in the absence and in the presence of NAD+. Human and murine PARP2 are highly homologous [32] and are expected to be indistinguishable in most functional aspects”. Taking into account that most of the functionally different proteins (PARP1, RPA, Ku 70/80, LigI) are human as well as the high homology between the human and murine PARP2s, we believe that the results are also valid for the fully human DNA repair system.

4. Additional experiments to strengthen the results of the paper. To my knowledge, there are no reports that suggest how Cas9 is ultimately removed from the DNA target after cleavage. The authors describe in the Discussion the high affinity for the DNA target and slow removal of Cas9. Is it possible that Cas9 is removed in an MRN-dependent manner, as Spo11 is removed from DSBs during meiosis in yeast and humans? This experiment could strengthen the results of this work as it could provide a mechanism in which Cas9 is ultimately removed.

We certainly agree with the Reviewer that the issue of Cas9 turnover in the cells is barely touched. Two ways that have found some experimental support so far involve FACT, a nucleosome disassembly factor that normally operates on H2A/H2B histones removing them from chromatin, and ubiquitylation or sumoylation followed by proteolysis [Wang et al., Mol. Cell. 2020;79:221-233; Ergünay et al., Life Sci. Alliance 2022;5:e202101078]. Even with this, the exact mechanism of Cas9 removal has not been addressed, and many other chromatin remodeling factors could be involved. The Reviewer’s suggestion about MRN is interesting; we also surmise that replication-coupled proteases resolving DNA-protein cross-links (SPRTN1, FAM111A, or DDI1) could be engaged. This topic is definitely worth investigating; however, it is so extensive that it cannot be addressed with a few additional experiments in a time frame of a paper revision. We plan to pursue this issue in the future; meanwhile, we have edited the text (p. 4, lines 68-72) to comment on the existing reports about Cas9 removal in cellulo.

Minor concerns:

1. The paper is well-written, but there are a few grammatical errors, throughout, including:

• Line 70, “DSB” should be “a DSB”

• Line 95 “taken” should be “used”

• Line 99 “we have then” should be “we then”

• Line 276 “SSB” should be “an SSB”

• Line 300, “except for the S and G2/M phases of the cell cycle” should be “except during the S and G2/M phases of the cell cycle”

• Line 471-472 “SSBs may be not easily drawn into” should be “SSBs may not be easily drawn into”

All the grammatical errors indicated have been corrected. 

2. Clarifying rationale/approach in a few experiments, including:

• Fig. 1, why is the plasmid incubated for only 2 min, but the dsDNA substrate for up to 30 min? Why are there multiple time points for the plasmid assays in panels C and D? It was helpful in Line 96 to provide the basal level of cleavage of the dsDNA substrate. Could this also be provided for the plasmid assay? For Fig 1 legend, please clarify by labeling panel C and D that the pLK1 plasmid was used in either the figure itself or the figure legend.

The incubation time for each substrate was chosen in preliminary experiments to achieve the extent of Cas9-induced cleavage within 30-60%, to be able to detect any possible modulation of the activity (either inhibition or activation). The extent of cleavage of the dsDNA substrate detected after a 30-min incubation did not exceed 42% and was almost maximal (corresponding to the plateau value). The extent of cleavage of the plasmid substrate detected even after a 1-min incubation was ~55% and further increased up to 80% after a 2-min incubation. The data therefore demonstrate different specific activities of Cas9 on the two types of DNA substrate.

The dsDNA substrate contained no detectable self-cleavage products. This is evident from the data presented in Fig. 1 A and B (control lane 1 in each panel). In the case of the plasmid substrate, the extent of DNA self-cleavage with generation of products co-migrating with the Cas9-induced cleavage products did not exceed 6% in the initial substrate and 10% after a 30-min incubation. This information has been added to the revised manuscript, at the end of the “Cas9 activity assay” subsection.

To clarify the type of DNA substrate in all experiments performed with the plasmid DNA, “S” in Figures 1 (C and D), 6 (A, B and C), and 7 (panel C) has been changed to “S (pLK1)”.

•Define dCas9 earlier in the results section. The first time it is introduced in Fig. S4, the “dCas9” should be defined as the double mutant. I had to look for this in the methods to determine what it meant.

We have defined dCas9 at the beginning of the second subsection of Results (page 6, lines 130-131): …..dCas9, the double D10A/H840A mutant fully inactive in DNA cleavage) “

• In Figure 2, line 168, the authors describe the mobility shift in the pLK1 plasmid as non-specific binding to the plasmid. It would be helpful to demonstrate this by repeating the experiment with an pLK1 plasmid without the dsDNA target sequence.

Nonspecific binding of Cas9 (either with or without sgRNA) to long DNA substrates was demonstrated many times by single-molecule visualization techniques [e. g., Sternberg et al., Nature 2014;507:62-67; Globyte et al., EMBO J. 2019;38:e99466; Ivanov et al., Proc. Natl Acad. Sci. USA 2020;117:5853-5860; Yang et al., Chem. Sci. 2021;12:12776-12784]. Nonspecific interaction of PARP1 and PARP2 with multiple undamaged binding sites in long DNAs was shown by atomic force microscopy [Sukhanova et al., Nucleic Acids Res. 2016;44:e60]. The results of these studies indicate independence of nonspecific binding on the DNA sequence. Taken into account the literature data, we have modified the description of results presented in Figure 2C as follows:

The previous text: “While the addition of increasing concentrations of free dCas9 induced a shift of the band of plasmid DNA, dCas9/sgRNA had no effect (Fig 2C). The dCas-induced band shift evidently reflects nonspecific binding of many protein molecules at multiple sites in the plasmid. Cas9 can bind DNA nonspecifically, although more than two orders of magnitude weaker than Cas9/sgRNA complex binding its target DNA [19]. On the other hand, dCas9/sgRNA binds a single site and thus contributes little (~190 kDa overall) to the total molecular weight of the plasmid (~3000 kDa), causing a negligible band shift. Nonspecific pLK1 binding by PARP1 and PARP2 was detected as a band shift at 100 nM and 500 nM, respectively (S6 Fig), reflecting higher affinity of PARP1 for the supercoiled DNA structure. The PARP1-induced band shift was not affected by addition of sgRNA but the band was supershifted to different extents upon addition of dCas9/sgRNA or dCas9 (Fig 2C), suggesting simultaneous interaction of the proteins with plasmid DNA. These results corroborate the hypothesis that PARPs do not interfere with the formation of a productive Cas9/sgRNA DNA complex.”

The new text: “While the addition of increasing concentrations of free dCas9 induced a shift of the band of plasmid DNA, dCas9/sgRNA had no effect (Fig 2C, lanes 14-16 vs. lanes 17-19). The dCas-induced band shift evidently reflects nonspecific binding of many protein molecules at multiple sites in the plasmid. Previously, Cas9 was shown to bind long DNA in a sequence-independent manner, although more than two orders of magnitude weaker in comparison with the site-specific binding of Cas9/sgRNA complex to its target DNA [19]. On the other hand, the specific binding of dCas9/sgRNA to a single site contributes little (~190 kDa overall) to the total molecular weight of the plasmid (~3000 kDa), causing a negligible band shift. Nonspecific pLK1 binding by PARP1 and PARP2 was detected as a band shift at 100 nM and 500 nM, respectively (S6 Fig), reflecting higher affinity of PARP1 for the undamaged DNA structure. Indeed, PARP1 was shown to exceed PARP2 by ~5-fold in the strength of nonspecific interaction with different DNAs [34]. The PARP1-induced band shift was not affected by addition of sgRNA but the band was supershifted to different extents upon addition of dCas9/sgRNA or dCas9 (Fig 2C), suggesting simultaneous interaction of the proteins with plasmid DNA. These results corroborate the hypothesis that PARPs do not interfere with the formation of a productive Cas9/sgRNA DNA complex.”

The legend to Figure 2 has also been modified:

The previous text: “Fig 2. Cas9/sgRNA, PARP1 and PARP2 binding to DNA. dsDNA1/2* (10 nM) was incubated in the absence of Mg2+ with Cas9/sgRNA (10 nM) in the absence or in the presence of PARP1 (A) or PARP2 (B), without or with 500 µM NAD+, and analysed by EMSA. In Panel C, pLK1 (10 ng/µl) was incubated in the absence or in the presence of PARP1 (200 nM) with the indicated amounts of dCas9/sgRNA, dCas9 alone or sgRNA alone. The sizes of DNA markers are shown next to the gel image.”

The new text: “Fig 2. Cas9/sgRNA, PARP1 and PARP2 binding to DNA. dsDNA1/2* (10 nM) was incubated with Cas9/sgRNA (10 nM) in the absence or in the presence of PARP1 (A) or PARP2 (B), without or with 500 µM NAD+, and analysed by EMSA. In Panel C, pLK1 (10 ng/µl) was incubated with the indicated amounts of dCas9/sgRNA, dCas9 alone or sgRNA (in the presence of Mg2+), in the absence or in the presence of PARP1 (200 nM). Positions of free (unbound) DNA or sgRNA and of various complexes are shown next to the respective gel images.”

• Figure 4, what do the numbers on the X axis of both the bar graphs mean? Do they correspond to the lane numbers on the respective gels? If yes, please add this information in the figure legend. If no, please remove, as they are a bit confusing.

We agree with this comment. The differently colored bars are specified in each panel of Figure 4. We have removed the numbers from the X axis.

• Figure 5, similar to Fig. 4, what do the numbers on the x axis of the bar graph mean? I don’t think the correspond to lane numbers of the corresponding gel, so they should probably be removed. 

The numbers on the X axis do not correspond to the lane numbers and have been removed.

• For the LIGI experiment, I’m a bit confused about the rationale for the experiment. If LIGI engages with a broken DNA molecule, why would it compete with Cas9 at a SSB? Wouldn’t Cas9 cleavage need to be completed before LIGI would bind? Or is the rationale that Cas9 may create a SSB that competes with LIGI and prevents a complete cleavage (DSB) reaction? If yes, clarifying this may strengthen the proposed rationale for the experiment.

DNA ligase I catalyses the ligation of SSB-containing DNA with a higher efficiency, similarly to LigIII, but in contrast to LigIV, which is more active in the ligation of DSB-containing DNA. Potentially, LigI can displace Cas9 nickase from the product and modulate the cleavage reaction kinetic. We show that LigI present together with nCas9 (H840A or D10A) has no effect on the cleavage reaction. Yet the addition of LigI to the cleavage reaction after thermal inactivation of nCas9 results in resealing of the cleavage product. To clarify these points, we have modified description of data presented in Fig. 6A and B as follows:

The previous text: “DNA ligases I and III are important DNA repair enzymes involved in the final repair step, the ligation of the nicked DNA. Potentially, they can compete with exogenous Cas9 nickases for the interaction with SSB. We have explored the influence of LigI on DNA cleavage by Cas9 nickases, using the pLK1 plasmid as a substrate to prevent additional interaction of LigI with blunt DNA ends. The addition of a 40-fold molar excess of LigI had no significant effect on the initial rate and maximal extent of the non-target strand cleavage by either nCas9 H840A/sgRNA or nCas9 D10A/sgRNA (Fig 6A and S8 Fig). After thermal inactivation of either nCas9, the cleaved DNA was nearly completely ligated (Fig 6B). However, the amount of cleaved DNA in the samples not subjected to heat treatment remained unchanged upon the addition of LigI. Hence, stable binding of nCas9 to their SSB products prevents the processing of SSBs by LigI.

Fig 6. Effects of SSB (DNA ligase I) and DSB (Ku70/80) repair sensors on the nickase activity of nCas9. (A) nCas9 H840A/sgRNA (10 nМ) was incubated with pLK1 (10 ng/µl) and LigI (400 nM) as indicated. The curves show the time course of cleaved product (P1) appearance (mean ± SD, n = 3). (B) nCas9 D10A/sgRNA or nCas9 H840A/sgRNA (10 nМ) was incubated with pLK1 (10 ng/µl) and LigI (400 nM) as indicated for 20 min. The samples marked with an asterisk were heat-inactivated (70°C for 5 min) before LigI addition. Bar charts show relative yields of the cleavage product under various reaction conditions (as specified in the legend) normalized to the respective value for each nickase without LigI and heating (mean ± SD, n = 3). The sizes of DNA markers are indicated next to the gel images. (C) Cas9/sgRNA (5 nM) was incubated with pLK1 (10 ng/µl) and Ku70/80 (100 nM) as indicated. The samples marked with an asterisk were pre-incubated for 10 min with Ku70/80 before the Cas9/sgRNA addition. The sizes of DNA markers are indicated next to the gel image. The curves show the time course of cleaved product appearance (mean ± SD, n = 3).”

The new text: “Eukaryotic DNA ligases I, III and IV are important DNA repair enzymes catalyzing, with different efficiencies, the ligation of DNAs containing SSBs or DSBs [46]. LigI can potentially modulate the nCas9-induced cleavage due to its high affinity for SSBs. We have explored the influence of human LigI on DNA cleavage by Cas9 nickases, using the pLK1 plasmid as a substrate to prevent additional low-affinity interaction of LigI with blunt DNA ends. The presence of a 40-fold molar excess of LigI in the cleavage reaction mixture had no significant effect on the initial rate and maximal extent of the non-target strand cleavage by either nCas9 H840A/sgRNA or nCas9 D10A/sgRNA (Fig 6A and S8 Fig). The cleavage product generated by either nCas9 was nearly completely ligated by LigI added after thermal inactivation of the nickase (Fig 6B). However, the amount of cleaved DNA in the samples not subjected to heat treatment remained unchanged upon the addition of LigI. Hence, stable binding of either nCas9 to its SSB product prevents the processing of SSB by LigI.

Fig 6. Effects of SSB (DNA ligase I) and DSB (Ku70/80) repair sensors on the nickase activity of nCas9. (A) pLK1 (10 ng/µl) was incubated with nCas9 H840A/sgRNA (10 nМ) and/or LigI (400 nM) as indicated. The curves show the time course of cleaved product (P1) appearance (the mean ± SD, n = 3). (B) pLK1 (10 ng/µl) was incubated with nCas9 D10A/sgRNA or nCas9 H840A/sgRNA (10 nМ), and/or LigI (400 nM) for 20 min. The samples marked with an asterisk were heat-inactivated (70°C for 5 min) before LigI addition and further incubated for 20 min. Bar charts show relative yields of the cleavage product under various reaction conditions (as specified in the legend) normalized to the respective value for each nickase without LigI and heating (the mean ± SD, n = 3). The sizes of DNA markers are indicated next to the gel images. (C) pLK1 (10 ng/µl) was incubated with Cas9/sgRNA (5 nM) and/or Ku70/80 (100 nM) as indicated. The samples marked with an asterisk were pre-incubated for 10 min with Ku70/80 before the Cas9/sgRNA addition. The sizes of DNA markers are indicated next to the gel image. The curves show the time course of cleaved product appearance (the mean ± SD, n = 3).”

3. Other minor concerns:

• Line 165, the authors mention the impact at 100 mM, but also a greater shift at 500 nM of PARP1 is apparent

A greater band shift at 500 nM PARP1 compared to 100 nM results from the further increase in the number of PARP1 molecules bound. In the text you mention, we only noticed that PARP1-induced band shift was detected at 5-fold lower concentration compared to that of PARP2.

• For the TIDE experiment, I do not believe this detects HR repair since an HDR substrate was not transfected. This should be mentioned in the Discussion or interpretation of the results.

We certainly agree that our TIDE experiments mostly detect NHEJ events (either c-NHEJ or alt-NHEJ) rather than HR. We have added a sentence “Since no homologous recombination template was provided, the editing reflects the repair of Cas9-mediated breaks by NHEJ” to the respective section of the Results (p. 17, lines 401-403).

• Line 622, how much of each vector was transfected and how many cells were transfected? This is particularly important if the transfection efficiency impacts DSB formation (and subsequent repair).

5×105 cells/well were seeded in a six-well plate and transfected with 2.5 µg of the plasmid after 24 h. This information has now been added to the Methods section (p. 26, lines 647-648).

• Fig. 9 includes three biological replicates, but the panel A only provides one sequencing read. For data sharing purposes, the other sequences should be provided, or in the very least, the figure legend should state that these are representative chromatograms from three biological replicates.

We have attached sequencing chromatograms from all experiments as S1 Appendix. In addition, we have edited the legend to Fig. 9 to indicate that it shows representative chromatograms.

Reviewer #2:

This manuscript by Maltseva and colleagues investigates how different factors involved in DNA repair or other DNA/RNA transaction mechanisms could potentially influence Cas9 behavior, both in terms of DNA binding and cleavage activity. The manuscript is clear and the results that are presented appear overall of good quality. Nevertheless, the relevance of some of the findings is questionable. This is particularly the case of the results regarding Cas9 ability to bind PAR as well as PARP2-dependent APD-ribosylation of Cas9, which appear somehow out of context and do not bring much to the current study. I would then suggest to remove these data from the manuscript. Regarding the other findings, I advise the authors to address the following concerns in order to consolidate their main conclusions.

PARP2-catalysed modification of Cas9 was shown here for the first time. The absence of PARP2 effects on the Cas9 activity may result from a low level of Cas9 modification under our reaction conditions. However, the biological role of this posttranslational modification cannot be excluded. We therefore believe that these data should remain in the paper.

1) The different results obtained in this manuscript rely on a single target DNA sequence and sgRNA. It would be interesting to control that, at least for a set of key experiments, similar results can be obtained with a different target sequence and sgRNA.

To explore possible interplay between Cas9 and DNA repair systems we use canonical and noncanonical DNA repair factors that interact with SSB/DSB (PARP1, PARP2, LigI, Ku70/80) or the DNA backbone (YB1) in a sequence-independent manner. Moreover, the issue of Cas9 sequence specificity mainly arises in the context of cleavage of non-targets rather than post-cleavage events at perfect targets. Thus, we deem it unlikely that our results will be significantly influenced by the target DNA sequence.

2) In several instances such as with PARP1 and Ku, the authors aim to look at a possible competition between Cas9 and the repair factors for binding at the DSB or nick generated by the Cas9. However, with the dsDNA, the results are most probably blurred by the ability of PARP1 and Ku to bind to the ends of the oligo. The use of the pLK1 plasmid is not very helpful here since the large size of this plasmid does not directly allows to assess protein binding by gel shift. It would be important to investigate further a direct competition between Cas9 and PARP1 or Ku at the Cas9-generated nick or DSB by using for example dumbbell DNA oligos that would prevent PARP1 or Ku binding at the DNA ends and also be small enough to be able to assess protein binding by gel shift assay.

The main goal of our study was to find protein factors among the major SSB/DSB sensors that would be capable of modulating Cas9 activity due to the displacement of the enzyme from its stable complex with the product. The dsDNA and plasmid DNA substrates of Cas9 were used primarily to test the catalytic activity of Cas9 and its nickase mutants in the absence or in the presence of different factors interacting with SSBs/DSBs. Binding experiments were performed to address one possible cause under the observed lack of detectable influence of the repair factors on the Cas9 activity. The only limitation of usefulness of the plasmid DNA is very small band shift upon protein binding at the single specific site. Yet site-specific binding of PARP1 (in the absence of Cas9) is evident from the dependence of its automodification level on the presence of SSB/DSB in pLK1 (Fig. 5 and its description). The data obtained in the binding experiments with the plasmid DNA are in full agreement with the results previously published for Cas9 and PARP1/PARP2 by others. We have added this information in the revised text (p. 8, lines 168-171 and 174-175). We doubt that dumbbell DNA will be more useful than the combined analysis of dsDNA and plasmid DNA. Moreover, the usefulness of dumbbell DNA is limited to SSB sensors.

3) On the gels shown on figure S4, the authors should further comment on the existence of two separated bands in the DNA bound fraction.

Two types of complexes with different gel mobility were detected only in the experiments with Cas9-sgRNA (Fig. 2, S4 Fig. and S9 Fig). This most probably results from different types of sgRNA folding in the complex. More than two types of Cas9-sgRNA complexes with dsDNA were observed in similar experiments by Doudna’s lab [Sternberg et al., Nature 2014;507:62-67].

4) On Fig 5, the author investigate how Cas9 mediated DSB on the pLK1 plasmid can induce PARP1 ADP-ribosylation. Since SSB, rather than DSB, is the primary substrate of PARP1, it would be interesting to perform the same experiment using the nickase version of the Cas9.

We note that preferential activation of PARP1 via binding to the DNA DSB (blunt ends or single-base overhangs) than to SSB was shown previously [D’Silva et al., Biochim. Biophys. Acta 1999;1430:119-126; Lilyestrom et al., J. Mol. Biol. 2010;395:983–994]. Actually, PARP1 is now believed to be the earliest DSB sensor that attracts the MRN complex to trigger ATM activation at a DSB [Wang et al., Nucleic Acids Res. 2006;34:6170-6182; Haince et al., J. Biol. Chem. 2008;283:1197-1208; Yang et al., Nucleic Acids Res. 2018;46:3446-3457; Krastev et al., Nat. Cell Biol. 2022;24:62-73]. Data presented in Fig. 5 demonstrate that PARP1 can be activated via interaction with supercoiled pLK1, but much less efficiently compared to the EcoRI-linearized plasmid. Similar data were obtained by using pBR plasmid in our previous study [Sukhanova et al., Nucleic Acids Res. 2016;44:e60]. Stimulation of PARP1 activity by Cas9-generated DSB depends on the product release induced by heating (Fig. 5). Data presented in S3 Fig demonstrate no influence of PARP1 on the activity of Cas9 nickase mutants, suggesting inability of PARP1 to displace the nickases from SSB.

5) On Fig 7C, the authors show that RPA is unable to stimulate Cas9 activity, in contrast to what they observed with the DNA duplex. The authors should comment further this difference between the two assays.

We have added the following sentence to the revised text (p. 14, lines 337-339): “This may be due to greater stability of the supercoiled plasmid DNA compared to the DNA duplex, which in turn makes DNA unwinding with RPA much more difficult.”

6) Information regarding sample size and other relevant statistics are generally missing and should be included.

The sample sizes in each experiment have been specified in the revised manuscript. All the experimental data were analysed by Student’s t-test to reveal statistically significant differences. The following information has been added to section Methods (p. 25, lines 605-606; p. 26, lines 640-642): “All the experiments were performed at least three times. The data were analysed for statistically significant differences by Student’s t-test”. Statistically significant differences are now indicated in the revised Figures 4, 5 and 7.

The references are modified in accordance with changes in the main text. 

The whole text of manuscript has been checked for errors.

Our original gel images are attached in Supporting Information as S1_raw_images.

The revised version of the manuscript has been approved by all authors. 

We would like to express our thanks to the Reviewers for their helpful comments. We hope that the revised version of our manuscript will be suitable for publication in PLoS ONE.

Sincerely yours,

Olga Lavrik and co-authors

---

## [Decision Letter · Decision Letter 1]

21 Aug 2023

PONE-D-23-20117R1Cas9 is mostly orthogonal to human systems of DNA break sensing and repairPLOS ONE

Dear Dr. Lavrik,

Thank you for submitting your revised manuscript to PLOS ONE. Both reviewers have re-examined the manuscript. However, reviewer 1 remains unconvinced about two important points that were raised during the initial evaluation (see detailed comments below). Therefore, we invite you to submit a revised version of the manuscript that addresses the points raised during the review process.

We look forward to receiving your revised manuscript.

Kind regards,

Zhiming Li, Ph.D.

Academic Editor

PLOS ONE

Journal Requirements:

Reviewers' comments:

Reviewer's Responses to Questions

**Comments to the Author**

1. If the authors have adequately addressed your comments raised in a previous round of review and you feel that this manuscript is now acceptable for publication, you may indicate that here to bypass the “Comments to the Author” section, enter your conflict of interest statement in the “Confidential to Editor” section, and submit your "Accept" recommendation.

Reviewer #1: All comments have been addressed

Reviewer #2: (No Response)

2. Is the manuscript technically sound, and do the data support the conclusions?

Reviewer #1: Yes

Reviewer #2: Partly

3. Has the statistical analysis been performed appropriately and rigorously? 

Reviewer #1: Yes

Reviewer #2: Yes

4. Have the authors made all data underlying the findings in their manuscript fully available?

Reviewer #1: Yes

Reviewer #2: Yes

5. Is the manuscript presented in an intelligible fashion and written in standard English?

Reviewer #1: Yes

Reviewer #2: Yes

6. Review Comments to the Author

Reviewer #1: The authors have addressed all of my major and minor concerns that are within the scope of this work. I believe the revisions provide clarity, stronger rationale, and with statistical analyses included, strengthen the rigor of their conclusions.

Reviewer #2: In this revised version of the manuscript by Maltseva et al., the authors addressed some of my comments. Nevertheless, I was less convinced by their answers to my first two concerns, which were actually the most critical ones.

In their response to my first concern, the authors do not provide any data to support their claim that "we deem it unlikely that our results will be significantly influenced by the target DNA sequence." As such, I feel that the fact that the authors did not test other target sequence and sgRNA is a major weakness of the current work.

Regarding my second point, unless I missed something, I still believe that the current competition experiments using dsDNA to test whether PARP1 or Ku could displace Cas9 and then affect its catalytic activity are difficult to interpret. Indeed both PARP1 and Ku will probably bind to the DNA ends with high affinity and therefore have no reason to compete with Cas9 since they bind to different locations along the dsDNA. This limitation is actually even mentioned by the authors themselves when they write that "To exclude binding of PARPs to the DNA duplex blunt ends, we further utilized the 163 supercoiled pLK1 plasmid as a DNA ligand." (line 162). Unfortunately, the use of the pLK1 plasmid is not completely appropriate since it does not allow to assess protein binding by gel shift. So overall I still feel that with the current samples (dsDNA and pLK1 plasmid) the authors are not able to adequately address the question of the competition between Cas9 and PARP1 or Ku.

7. PLOS authors have the option to publish the peer review history of their article (what does this mean?). If published, this will include your full peer review and any attached files.

Reviewer #1: No

Reviewer #2: No

---

## [Author Response · Author response to Decision Letter 1]

4 Nov 2023

Dear Reviewers,

Thank you for considering the reviewed version of our manuscript, “Cas9 is mostly orthogonal to human systems of DNA break sensing and repair” (PONE-D-23-20117R1) by Maltseva et al. As Reviewer 1 did not raise further concerns, below please find our response to the remaining questions from Reviewer 2.

Reviewer #2: In this revised version of the manuscript by Maltseva et al., the authors addressed some of my comments. Nevertheless, I was less convinced by their answers to my first two concerns, which were actually the most critical ones.

In their response to my first concern, the authors do not provide any data to support their claim that "we deem it unlikely that our results will be significantly influenced by the target DNA sequence." As such, I feel that the fact that the authors did not test other target sequence and sgRNA is a major weakness of the current work.

We certainly agree that Cas9 can be targeted by sgRNA to different sequences. The main point of our initial response was that PARP1, PARP2, LigI, Ku70/80 and YB1 are non-sequence-specific proteins, and it can hardly be expected that they will interact with other sequences in a manner different from the Sp2 sequence. To check possible influence of the target DNA sequence on the activity of Cas9 and its interplay with key DNA repair factors, we performed additional experiments with another sgRNA and target DNA plasmid substrate.

The sequences of a new sgRNApMSH2 and the previously used sgRNApLK1 are presented below:

sgRNApMSH2

5′ AUCAAGUACAUGGGGCCGGCGUUUUAGAGCUAGAAAUAGCAAGUUAAAAUAAGGCUAGUCCGUUAUCAACUUGAAAAAGUGGCACCGAGUCGGUGCUUUU-3′

sgRNApLK1

5′ GGAUAACUCAAUUUGUAAAAAAGUUUUAGAGCUAGAAAUAGCAAGUUAAAAUAAGGCUAGUCCGUUAUCAACUUGAAAAAGUGGCACCGAGUCGGUGCUUUU-3′.

The DNA plasmid substrate corresponding to sgRNApMSH2 was constructed based on a plasmid with inserted hMSH2 gene (Addgene ID #16453).

First, we compared the Cas9-catalysed cleavage of both substrates (Fig. 1).

Fig 1. Cleavage of two different DNA plasmid substrates by Cas9/sgRNA. Cas9/sgRNApLK1 (20 nM) or Cas9/sgRNApMSH2 (20 nM) was incubated with pLK1 DNA (10 ng/µl) or pMSH2 DNA (10 ng/µl), respectively. S, substrate (supercoiled plasmid pMSH2 or pLK1); P1, SSB containing product (nicked plasmids); P2, DSB-containing product (linear plasmids). The curves show the accumulation of P1+P2 products under the indicated conditions. The sizes of DNA markers are shown next to the gel images.

The activity of Cas9 in cleavage of the pMSH2 DNA was revealed to be 3-fold lower compared the pLK1 DNA. The difference most likely results from the higher GC content of the protospacer seed region of the former substrate. 

Next, we explored the influence of several DNA binding proteins on the of Cas9-catalysed cleavage of the pMSH2 DNA (Fig. 2). No significant change in the efficiency of pMSH2 DNA cleavage was detected in the presence of PARP1/2 at a high excess over Cas9, regardless of the presence of NAD+. Similarly, no modulating effects produced by either Ku70/80 or RPA could be observed. Thus, the ability of Cas9 to efficiently shield DSBs and SSBs from their sensors does not depend on the targeted DNA sequence.

Fig 2. Cleavage of the pMSH2 plasmid substrate by Cas9/sgRNApMSH2 in the presence of different proteins. Cas9/sgRNApMSH2 (20 nM) was incubated with pMSH2 DNA (10 ng/µl) in the absence and presence of PARP1 (100 nM) or PARP2 (500 nM) without or with 500 μM NAD+, Ku70/80 (250 nM), and RPA (500 nM) for 15 min at 37°C. S, substrate (supercoiled pMSH2 plasmid); P1, SSB containing product (nicked plasmid); P2, DSB-containing product (linear plasmid). The bar chart shows the extent of pMSH2 DNA cleavage in the presence of the indicated proteins.

All new data, including the supplementary figure S10, have been added to the revised text (p. 15, lines 353-359). According to this the numbering of supplementary figures and the supplementary figure references in main text have been changed. 

Regarding my second point, unless I missed something, I still believe that the current competition experiments using dsDNA to test whether PARP1 or Ku could displace Cas9 and then affect its catalytic activity are difficult to interpret. Indeed both PARP1 and Ku will probably bind to the DNA ends with high affinity and therefore have no reason to compete with Cas9 since they bind to different locations along the dsDNA. This limitation is actually even mentioned by the authors themselves when they write that "To exclude binding of PARPs to the DNA duplex blunt ends, we further utilized the supercoiled pLK1 plasmid as a DNA ligand." (line 162). Unfortunately, the use of the pLK1 plasmid is not completely appropriate since it does not allow to assess protein binding by gel shift. So overall I still feel that with the current samples (dsDNA and pLK1 plasmid) the authors are not able to adequately address the question of the competition between Cas9 and PARP1 or Ku.

We surmise that we and the Reviewer are actually talking about the same thing but our narrative was too confusing. The purpose of these experiments was exactly to see whether PARPs’ apparent lack of competition with Cas9 observed when following the activity is due to binding to different parts of DNA. Essentially, the ds oligo experiments is relevant to the lack of competition on dsDNA1/2, presented in Fig. 1A,B, whereas the plasmid experiment is relevant to the lack of competition on pLK1, presented in Fig. 1C,D. With the ds oligo, the lack of binding competition is clearly seen from the same PARP concentration dependence of band shift without (Supplementary Fig. S5) or with Cas9 (Fig. 2A,B). The plasmid experiment, on the other hand, tells us that even at the PARP:plasmid molar ratio of ~60:1 (200 nM) PARP1 molecules are bound along DNA in an amount significant enough to shift it (Supplementary Fig. S5, refer to lanes 12 and 13) but this has little positive or negative effect on Cas9 binding with or without sgRNA (Fig. 2C).

The reviewer’s suggestion to use alternative DNA structures does not seem particularly useful, since at least PARP1 and Ku70/80 are known to bind various non-canonical nucleic acid structures, including hairpin loops [Laspata et al., J. Mol. Biol., 2023, 20:168207. doi: 10.1016/j.jmb.2023.168207; Arosio et al., J. Biol. Chem., 2002, 277:9741-8. doi: 10.1074/jbc.M111916200]. Thus, such substrates would not help to eliminate the competition at DNA duplex ends, as a circular plasmid does.

The revised version of the manuscript has been approved by all authors. 

Sincerely yours,

Olga Lavrik and co-authors

---

## [Editor Report · Decision Letter 2]

7 Nov 2023

Cas9 is mostly orthogonal to human systems of DNA break sensing and repair

PONE-D-23-20117R2

Dear Dr. Lavrik,

Thank you for the efforts in addressing reviewers' comments. We’re pleased to inform you that your manuscript has been judged scientifically suitable for publication and will be formally accepted for publication once it meets all outstanding technical requirements.

Kind regards,

Zhiming Li, Ph.D.

Academic Editor

PLOS ONE

---

## [Editor Report · Acceptance letter]

17 Nov 2023

PONE-D-23-20117R2 

Cas9 is mostly orthogonal to human systems of DNA break sensing and repair 

Dear Dr. Lavrik:

I'm pleased to inform you that your manuscript has been deemed suitable for publication in PLOS ONE. Congratulations! Your manuscript is now with our production department. 

Kind regards, 

on behalf of

Dr. Zhiming Li 

Academic Editor

PLOS ONE